# Fission yeast Rad54 prevents intergenerational buildup of Rad51 aggregates in proliferating cells

Goki Taniguchi[1,2], Alexander I May[1,2] ⓘ, Hiroshi Iwasaki[1,2] ⓘ, Hideo Tsubouchi[1,2] ⓘ

**Homologous recombination is central to the maintenance of genome stability. Using fission yeast, we found that mutation of the *rad54* gene leads to robust Rad51 accumulation in vegetatively growing cells. By developing a protocol to track Rad51 in live yeast cells, we traced the origin and fate of Rad51 aggregates formed in *rad54* mutants. Our observations strongly suggest that DNA breaks arising in late S phase act as the primary initiators of Rad51 accumulation. Rad51 initially appears as foci during late S phase, which continue to enlarge throughout the G2 phase. These Rad51 accumulations frequently persist into M phase and are distributed along with chromosomes into daughter cells. The inherited Rad51 mass in daughter cells continues to grow, forming robust Rad51 aggregates that are often associated with cell cycle arrest. Thus, the primary role of Rad54 in vegetative fission yeast cells is to facilitate the repair of DNA breaks arising in late S phase. The intergenerational accumulation of Rad51 aggregates in *rad54* mutants reveals a novel mechanism through which defective homologous recombination drives genome instability.**

## Introduction

Homologous recombination (HR) plays a critical role in the maintenance of genome stability by identifying sequence similarities in DNA molecules, which allows for accurate repair of DNA breaks including double-strand breaks (DSBs) (San Filippo et al, 2008; Heyer et al, 2010; Mehta & Haber, 2014; Symington et al, 2014; Kowalczykowski, 2015). Defects in HR mechanisms induce elevated levels of genome instability that can ultimately give rise to cancers and certain genetic disorders/syndromes of high cancer incidence (San Filippo et al, 2008; Heyer et al, 2010; Prakash et al, 2015).

A key player in HR is the eukaryotic RecA-family homologous recombinase Rad51 (Brown & Bishop, 2015; Kowalczykowski, 2015). After the formation of a DSB, the newly exposed ends of the break are digested by a group of nucleases to produce 3′-tailed single-stranded DNA (ssDNA) (Cejka & Symington, 2021). ssDNA is then coated by the replication protein A complex (RPA) (Chen & Wold, 2014) before the proteins, collectively known as the recombination mediator, facilitate the replacement of RPA by Rad51 to promote the formation of the Rad51 nucleoprotein filament (Sung et al, 2003). This Rad51 filament enables the exposed ssDNA sequence to identify intact homologous double-stranded DNA (dsDNA) that can be used as a template for DSB repair. Rad51 facilitates the pairing of ssDNA with complementary sequences to form a displacement loop (D-loop) (San Filippo et al, 2008; Heyer et al, 2010; Mehta & Haber, 2014; Symington et al, 2014; Kowalczykowski, 2015). Repair DNA synthesis starts from the 3′ end of the invading strand, restoring lost information by using its complementary strand as a template.

The Swi2/Snf2 family DNA translocator Rad54 also plays several important roles in HR (Ceballos & Heyer, 2011). It facilitates formation of the Rad51 nucleoprotein filament (Mazin et al, 2003; Agarwal et al, 2011), strongly promotes Rad51-mediated strand exchange (Petukhova et al, 1998; Van Komen et al, 2000), and drives Holliday junction branch migration (Bugreev et al, 2006). In addition, Rad54 drives Rad51 nucleoprotein filament association with target duplex DNA (Tavares et al, 2019) and is also able to act as a heteroduplex DNA pump that drives D-loop formation and removes Rad51 as heteroduplex DNA is formed (Solinger et al, 2002; Wright & Heyer, 2014). In addition, single-molecule analyses have demonstrated that Rad54 translocates along dsDNA (Amitani et al, 2006) with the Rad51 nucleoprotein filament, promoting Rad51-mediated homology search through local unwinding of donor dsDNA (Crickard et al, 2020). Removal of Rad51 filaments associated with heteroduplex DNA by Rad54 enables the initiation of repair synthesis (Li & Heyer, 2009).

Rad51 is able to bind both dsDNA and ssDNA (Zaitseva et al, 1999; Tombline et al, 2002). Whereas binding to ssDNA allows Rad51 to be targeted to damaged DNA, nonspecific dsDNA biding can be problematic as most chromosomal DNA is intact and Rad51 filament formation on dsDNA is not only unproductive but also inhibits HR (Ogawa et al, 1993; Benson et al, 1994; Sung & Robberson, 1995). Rad54-family DNA translocases have been implicated in Rad51 removal from non-specifically bound dsDNA in budding yeast: Rad51 foci unassociated with DNA damage

[1]Cell Biology Center, Institute of Integrated Research, Yokohama, Japan    [2]School of Life Science and Technology, Institute of Science Tokyo, Yokohama, Japan

Correspondence: hiwasaki@bio.titech.ac.jp; htsubouchi@bio.titech.ac.jp

accumulate in the absence of Rad54, the Rad54-homolog Rdh54, and the third Swi2/Snf2 family translocase, Uls1 (Shah et al, 2010). During meiosis, Dmc1, the meiosis-specific Rad51 paralog, accumulates on undamaged chromosomes in the absence of Rdh54 (Holzen et al, 2006), and similar nonspecific Rad51 accumulation occurs following down-regulation of Rad54 in human cells (Mason et al, 2015).

Rdh54, a homolog of Rad54, is found in many eukaryotes (Ceballos & Heyer, 2011). While budding yeast Rdh54 shares similar biochemical properties with Rad54, it functions predominantly during meiosis (Klein, 1997; Shinohara et al, 1997). However, in budding yeast, Rdh54 plays nonessential roles in mitotic DSB repair, affecting the properties of the D-loop (Anand et al, 2014; Tsaponina & Haber, 2014; Shah et al, 2020; Sugawara et al, 2024). The fact that *Schizosaccharomyces pombe* Rdh54 functions exclusively in meiosis (Catlett & Forsburg, 2003) makes this organism an excellent model for studying the specific role of Rad54 in HR during vegetative growth.

Here, we demonstrate that in fission yeast, the loss of Rad54 induces massive, aberrant accumulation of Rad51 under normal growth conditions, ultimately resulting in cell cycle arrest. We have found that Rad51 accumulation begins during the late stage of S phase and is dependent on major Rad51 mediator factors including Rad52, Rad57, and Sfr1 (Tsubouchi et al, 2021), as well as Exo1 and Rqh1, which are components of the long-range resection machinery required for the production of ssDNA substrate for Rad51 (Yan et al, 2019; Cejka & Symington, 2021). Accumulating Rad51 is distributed with chromosomes into daughter cells. An inherited mass of Rad51, which we term a Rad51 body, progressively increases in size over successive generations, ultimately forming large aggregates that hinder cell cycle progression. Our results strongly suggest that Rad51 accumulation is initiated by exposed ssDNA arising from either single-strand breaks (SSBs) or DSBs, underlining the importance of Rad54 in promoting the repair of spontaneous late-S phase chromosomal breaks that otherwise initiate runaway intergenerational aggregation of Rad51.

# Results

## Loss of Rad54 causes massive accumulation of Rad51

Rad54 plays various important roles in HR (Ceballos & Heyer, 2011). We aimed to understand Rad54's primary role in the regulation of Rad51 in normally proliferating fission yeast cells. We deleted the *rad54* gene in haploid cells to derive a *rad54* mutant strain and examined the localization of Rad51 by immunostaining. Whereas wild type (WT) cells occasionally contained a subtle Rad51 signal within the nucleus, a substantial fraction of *rad54* mutant cells exhibited robust Rad51 accumulation (Fig 1A). We noticed that accumulated Rad51 exhibits a characteristic morphology, with the formation of both foci and fibers apparent. Quantification of Rad51 accumulation morphologies in individual cells revealed three categories: focus, short fiber, or long fiber (Fig 1B). In *rad54*

mutant cells, the majority (~50%) exhibited long fibers, whereas only foci were observed in WT cells at very low frequency (Fig 1C).

Recruitment of Rad51 to DSBs requires Rad51 auxiliary factor proteins (Symington et al, 2014; Wright et al, 2018; Tsubouchi et al, 2021), and Rad51 foci are thought to represent Rad51 nucleoprotein filaments created at the exposed ends of DSBs. In *S. pombe*, Rad52 and the Rad55-Rad57 and Swi5-Sfr1 complexes contribute to Rad51 foci formation (Akamatsu et al, 2007; Lorenz et al, 2009; Argunhan et al, 2020; Afshar et al, 2021). Each single mutant exhibited a small fraction of cells carrying Rad51 foci (~15%), comparable to the level observed in the WT. In the *rad54* mutant background, loss of Rad52 completely abolished the Rad51 fibers (Fig 1C and D). The absence of Rad57 or Sfr1 also caused a substantial reduction in accumulated Rad51 fibers. In contrast to the *sfr1* mutant, where only foci were observed, a considerable fraction of *rad57* mutant cells also exhibited short fibers (13%), suggesting that the impact of *rad57* mutation on Rad51 assembly is milder than the *sfr1* mutation (Akamatsu et al, 2007) (Fig 1C and D). Together, these results suggest that Rad51 localization and *rad54* mutant-dependent aggregate formation represent Rad51 accumulation on DNA as part of the DSB repair mechanism.

## Accumulation of Rad51 is inhibitory to cell growth

The formation of such striking aggregates on DNA suggests that the lack of Rad54 has a strong effect on cell physiology. Meanwhile, the accumulation of Rad51 on DNA is known to be associated with genome instability and has an inhibitory effect on cell growth (Shah et al, 2010; Mason et al, 2015; Muraszko et al, 2021). During microscopy analyses, we noticed that *rad54* mutant cells are significantly longer than WT cells (Fig 2A and B). Increased cell length often indicates a persistence of cells in the G2 phase of the cell cycle. Analyses of these data revealed a correlation between Rad51 signal intensity and cell length, especially in *rad54* mutant cells (Fig 2C; $R^2$ = 0.34 and 0.64 for WT and *rad54*, respectively). Interestingly, cell length data indicated a statistically significant increase in *rad54* mutant cell length in comparison to *rad51* cells, but double mutant *rad51 rad54* cells appeared to be a similar length to *rad51* mutants. We observed a similar effect when examining dissected tetrads, whereby the loss of Rad54 alone had the greatest effect on colony size (Fig 2D), as well as for the growth of cells as manifested in growth rates (Fig 2E and F). We also assessed the viability of vegetatively proliferating cells in these strains (Fig 2G). While both *rad51* and *rad54* mutants showed considerably lower cell viability, the *rad54* mutant was characterized by a more pronounced decrease than *rad51*, with viabilities of ~30% for *rad54* and 50% for *rad51*. As with other physiological analyses, the *rad51 rad54* double mutant exhibited a viability similar to that of *rad51*, in contrast to the lower viability observed upon *rad54* mutation alone. The stronger physiological phenotypes observed in *rad54* cells than in *rad51* and *rad54 rad51* cells suggest that Rad51 accumulation in the absence of Rad54 results in more severe physiological implications for cells than the disruption of the DNA damage repair machinery caused by the loss of Rad51.

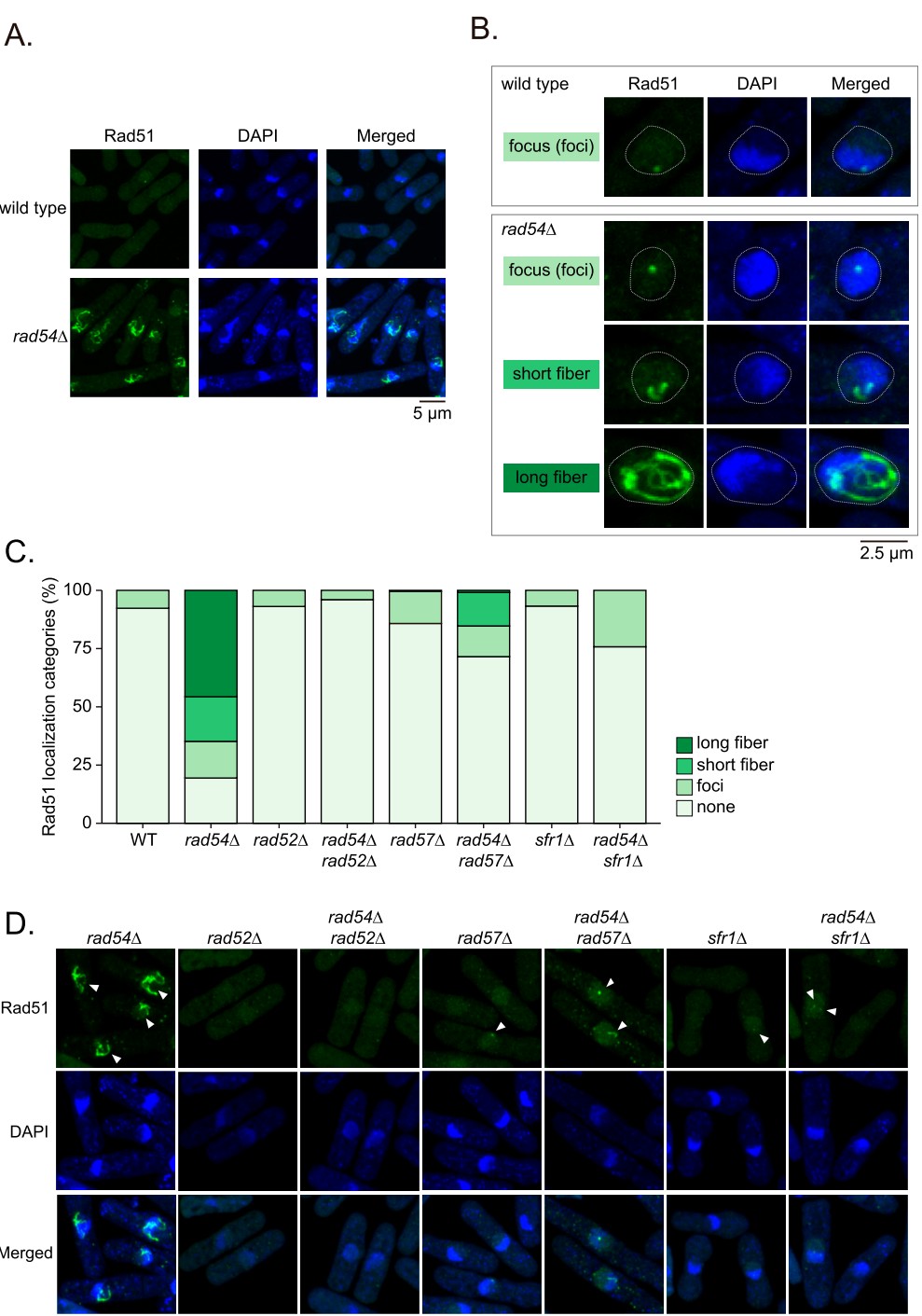

**Figure 1. Loss of Rad54 causes accumulation of Rad51.**
**(A)** Immunostaining of Rad51 and DNA (DAPI) in exponentially growing cells. **(B)** Morphological categorization of Rad51. Cells containing Rad51 fibers less than 2.5 µm in length categorized as "short," those with longer fibers categorized as "long." Cells at M phase and dead cells were excluded. **(C)** Frequency of Rad51 morphological categories in indicated strains. n: WT = 572; *rad54Δ* = 324; *rad52Δ* = 332; *rad54Δ rad52Δ* = 376; *rad57Δ* = 429; *rad54Δ rad57Δ* = 327; *sfr1Δ* = 444; *rad54Δ sfr1Δ* = 289. **(D)** Representative images showing Rad51 morphologies in indicated mutants. White arrowheads indicate Rad51 accumulations. WT, wild type. Strains used: WT (GT414), *rad54Δ* mutant (GT422), *rad52Δ* (GT110), *rad54Δ rad52Δ* (GT530), *rad57Δ* (BA269), *rad54Δ rad57Δ* (GT473), *sfr1Δ* (GT479), and *rad54Δ sfr1Δ* (GT476). Source data are available for this figure.

## The DNA damage checkpoint is constitutively activated in the absence of Rad54

*rad54* mutant cells are extremely elongated (Fig 2B). This abnormal morphology is reminiscent of cells responding to DNA stresses such as DNA damage or replication stress. We therefore hypothesized that so-called checkpoint surveillance mechanisms, which are responsible for sensing of DNA stress and delaying cell cycle progression until such stress is rectified (Hartwell & Weinert, 1989), may be involved. In *S. pombe*, DNA stress signals are mediated by the major effector kinases Chk1 and Cds1 (Russell, 1998; Carr, 2002; McGowan, 2002; Morgan & Canman, 2018). While Chk1 is primarily responsible for DNA damage signaling during G2 phase (damage checkpoint), Cds1 is

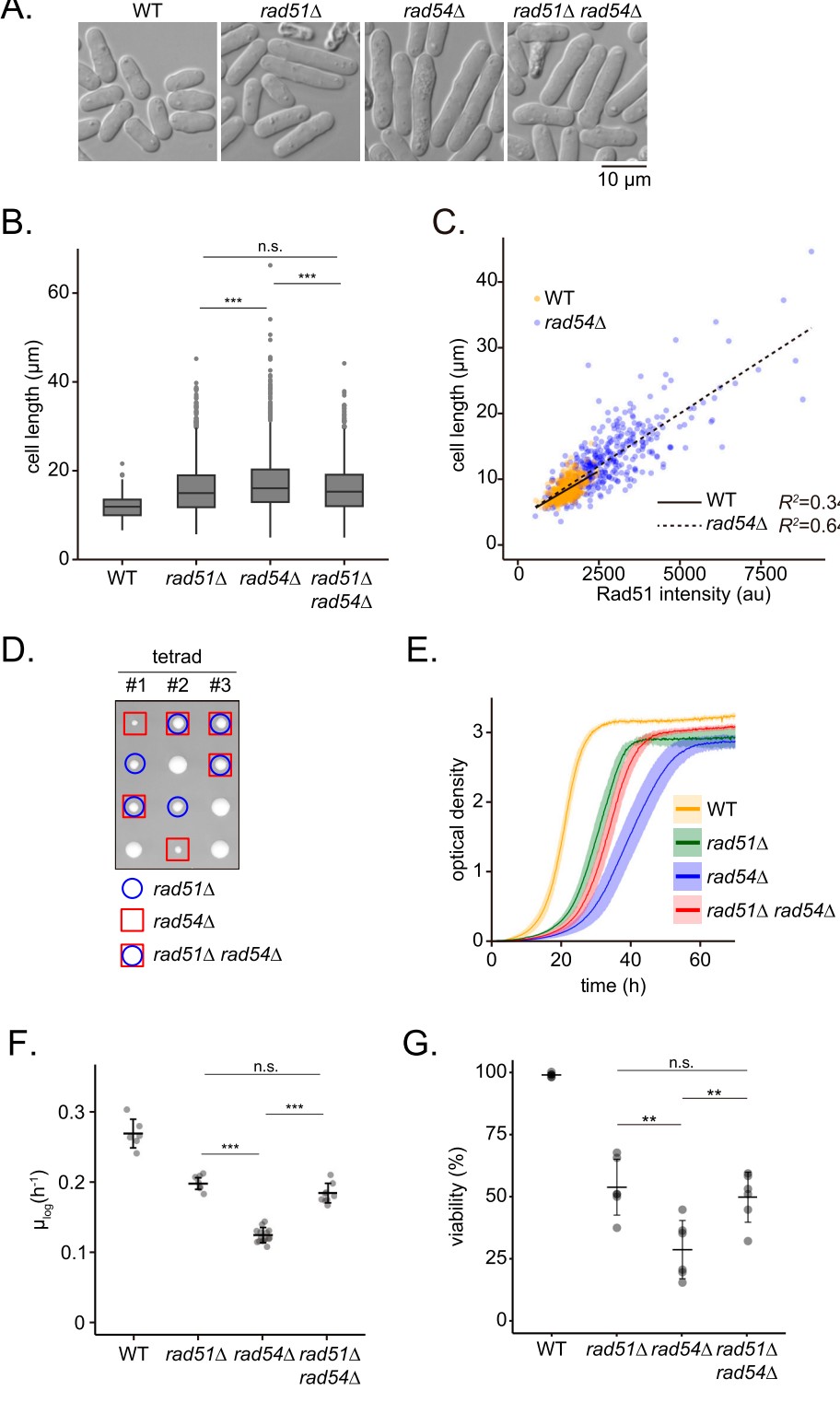

**Figure 2. Slow-growth of the *rad54* mutant is partially alleviated by the loss of Rad51.**
**(A)** Representative cell morphologies of indicated strains. **(B)** Quantitative analysis of cell length. Data points per strain were visualized as a box-and-whisker plot. Center line, median; box limits, upper and lower quartiles. Whiskers, 1.5 x IQR, with outliers indicated by gray dots. n: WT = 1,419; *rad51Δ* = 2,255; *rad54Δ* = 2,139; *rad51Δ rad54Δ* = 1,563. Statistical differences were analyzed by Dunn's test (*rad51Δ* versus *rad54Δ*, $P < 0.001$; *rad51Δ* versus *rad51Δ rad54Δ*, $P = 0.608$; *rad54Δ* versus *rad51Δ rad54Δ*, $P < 0.001$). **(C)** Relationship between Rad51 signal intensity and cell length. Measured total Rad51 signal per nucleus and cell length are shown. Solid and broken lines represent simple linear regression with the coefficient of determination ($R^2$) indicated. n: WT = 417; *rad54Δ* = 417. **(D)** Haploid colonies from the dissection of tetrads formed by crossing *rad51Δ rad54Δ* (GT426) and WT (GT416). **(E)** Growth of indicated strains. Solid lines represent averages of biological replicates, shaded areas SD (WT, n = 6; *rad51Δ*, n = 7; *rad54Δ*, n = 15; *rad51Δ rad54Δ*, n = 9). **(F)** Maximum growth rate ($\mu_{log}$) of indicated strains determined using data in (E). n: WT = 6; *rad51Δ* = 7; *rad54Δ* = 15; *rad51Δ rad54Δ* = 9. Data shown as mean ± SD. Tukey's HSD test was performed to compare mutant strains (*rad51Δ* versus *rad54Δ*, $P < 0.001$; *rad51Δ* versus *rad51Δ rad54Δ*, $P = 0.683$; *rad54Δ* versus *rad51Δ rad54Δ*, $P < 0.001$). **(G)** Viability of indicated strains. Cells were individually placed on a grid using a tetrad dissector, and the number of colonies was divided by the total number of cells examined. The experiment was repeated six times. Data shown as mean ± SD. Tukey's HSD test was performed to compare strains (*rad51Δ* versus *rad54Δ*, $P = 0.00105$; *rad51Δ* versus *rad51Δ rad54Δ*, $P = 0.886$; *rad54Δ* versus *rad51Δ rad54Δ*, $P = 0.00547$). Strains used: WT (GT414), *rad51Δ* (GT418), *rad54Δ* (GT422), and *rad51Δ rad54Δ* (GT426). WT, wild type.
Source data are available for this figure.

implicated in replication stress signaling during S phase (replication checkpoint) (Rhind & Russell, 2000). To determine whether Rad54 is implicated in either checkpoint, *cds1* or *chk1* mutation was introduced into *rad54* mutant cells and the effect on cell length assessed (Fig 3A). *rad54 cds1* mutant cells exhibited a similar morphology to that of the *rad54* single mutant, with their cell length substantially longer than WT. In contrast, *rad54 chk1* double mutant cells were much shorter than

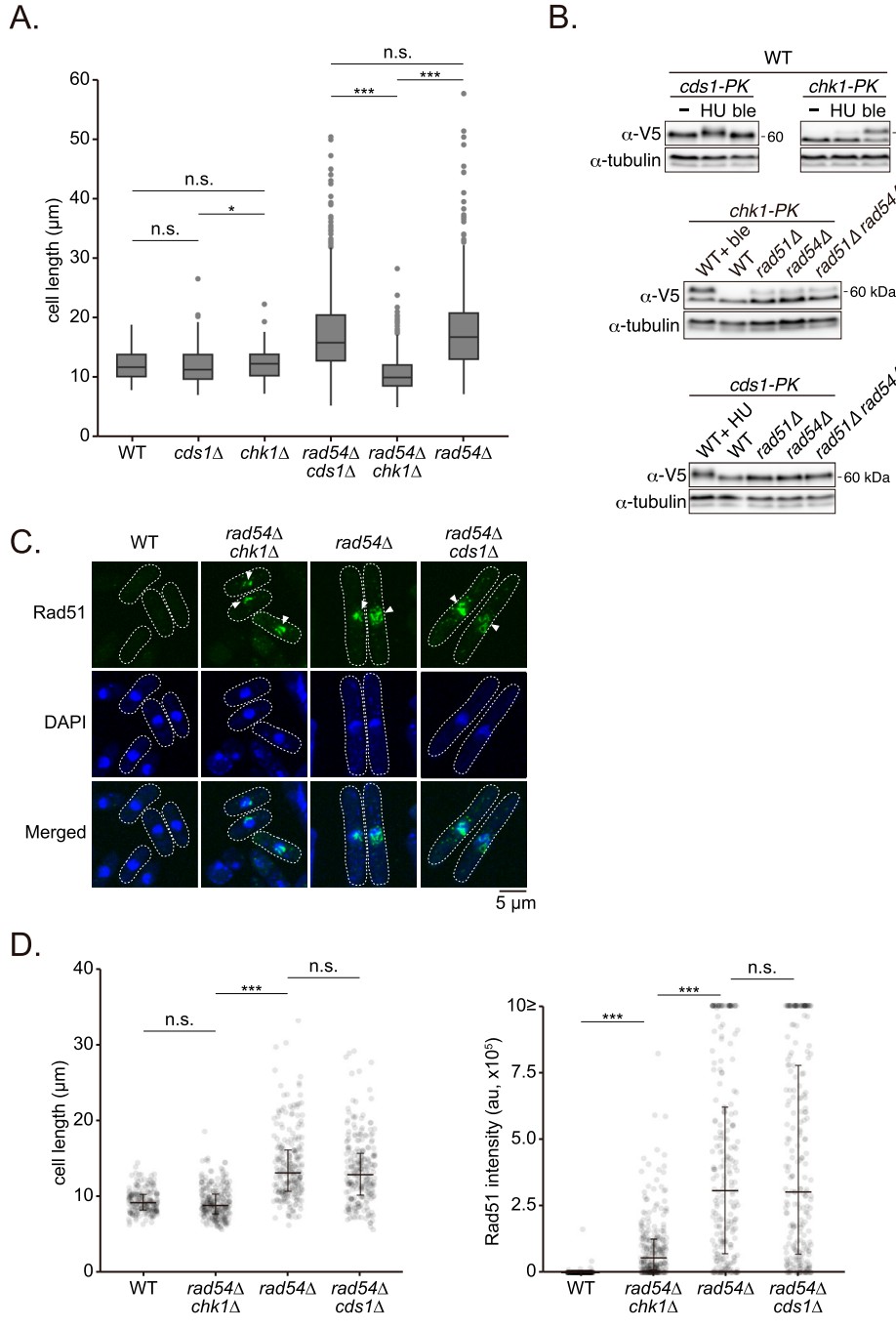

**Figure 3. The DNA damage checkpoint is constitutively active in *rad54* cells.**
**(A)** Quantitative analysis of cell length. Data are shown as box-and-whisker plots. Center line, median; box limits, upper and lower quartiles. n: WT = 965; *cds1Δ* = 929; *chk1Δ* = 702; *rad54Δ cds1Δ* = 915; *rad54Δ chk1Δ* = 820; *rad54Δ* = 845. Statistical significance was determined by Dunn's test (WT versus *cds1Δ*, P = 0.417; WT versus *chk1Δ*, P = 1; *cds1Δ* versus *chk1Δ*, P = 0.022; *rad54Δ* versus *rad54Δ cds1Δ*, P = 0.478; *rad54Δ* versus *rad54Δ chk1Δ*, P < 0.001, *rad54Δ cds1Δ* versus *rad54Δ chk1Δ*, P < 0.001). **(B)** PK-tagged Cds1 and Chk1 proteins were detected using anti-V5 antibody by Western blotting. Cells were treated with 20 mM HU or 1 μg/ml bleomycin for 2 h before harvesting. Tubulin was used as a loading control. WT, wild type; HU, hydroxyurea; ble, bleomycin. **(C)** Representative Rad51 localization in the indicated strains. Arrowheads indicate Rad51 aggregates. **(D)** Quantification of cell length and Rad51 fluorescence intensity in individual cells. au, arbitrary unit. Center line, median; line limits, upper and lower quartiles. n: WT = 201; *rad54Δ chk1Δ* = 265; *rad54Δ* = 237; *rad51Δ cds1Δ* = 239. For cell length, statistical differences were analyzed by Dunn's test (*rad54Δ* versus *rad54Δ cds1Δ*, P = 0.105; *rad54Δ* versus *rad54Δ chk1Δ*, P < 0.001; WT versus *rad54Δ chk1Δ*, P = 0.250). For Rad51 fluorescence intensity, statistical differences were analyzed by Dunn's test (*rad54Δ* versus *rad54Δ cds1Δ*, P = 0.840; *rad54Δ* versus *rad54Δ chk1Δ*, P < 0.001; WT versus *rad54Δ chk1Δ*, P < 0.001). Strains used: WT (GT414), *cds1Δ* (GT1254), *chk1Δ* (GT1257), *rad54Δ* (GT422), *rad54Δ cds1Δ* (GT1362), *rad54Δ chk1Δ* (GT1272), *cds1-PK* (GT1241), *cds1-PK rad51Δ* (GT1286), *cds1-PK rad54Δ* (GT1397), *cds1-PK rad54Δ rad51Δ* (GT1295), *chk1-PK* (GT1251), *chk1-PK rad51Δ* (GT1300), *chk1-PK rad54Δ* (GT1378), *chk1-PK rad54Δ rad51Δ* (GT1307).
Source data are available for this figure.

*rad54* cells and instead comparable to WT cells, suggesting that *rad54* cells activate the DNA damage checkpoint.

The activation of these checkpoint pathways can be monitored by phosphorylation of Cds1 and Chk1 (Walworth & Bernards, 1996; Lindsay et al, 1998). Cds1 and Chk1 were tagged with PK to assess their phosphorylation by gel mobility. Strains expressing Cds1-PK or Chk1-PK exhibited resistance to both bleomycin, a radiomimetic agent that causes DNA damage, and hydroxyurea (HU), which inhibits DNA replication, confirming that these fusion proteins are functional (Fig S1A). A clear Cds1-PK band shift was observed in

WT cells exposed to HU, while a very small fraction of Chk1-PK also exhibited a mobility shift under these conditions (Fig 3B). In contrast, bleomycin treatment resulted in reduced mobility of a marked fraction of Chk1-PK exclusively. Meanwhile, Chk1-PK mobility shift was detected without any drug treatment in *rad54*, *rad51* and *rad51 rad54* mutants, while band shift was not observed in mutant cells expressing Cds1-PK (Fig 3B).

We next examined the impact of checkpoint defects on the accumulation of Rad51 in *rad54* mutant strains by confocal microscopy. Substantial accumulation of Rad51 in all *rad54*

background strains was detected (Fig 3C). However, we noticed that in the *rad54 chk1* double mutant, cell length is equivalent to WT cells and the total amount of Rad51 signal per cell is significantly lower than that observed in *rad54* single or *rad54 cds1* double mutant strains (Fig 3D). The colony size of the *rad54 chk1* double mutant was found to be smaller than that of the *rad54* single mutant, thereby suggesting the synthetic effect between HR and checkpoint defects caused by these mutations (Fig S1B). In summary, the constitutive activation of Chk1 in the absence of Rad54 suggests that DNA damage, not replication stress, is responsible for delayed cell cycle progression. Furthernore, reduced Rad51 accumulation observed in the *rad54 chk1* double mutant indicates Rad51 accumulation primarily takes place during the extended G2 phase caused by the activation of the DNA damage checkpoint.

### Functional interaction between HR-related helicases and Rad54

While *rad54* mutant cells exhibit profound growth defects, a quarter of these cells is viable, indicating that alternative DNA repair mechanisms allow these cells to survive. In order to identify the basis for viability in the absence of Rad54, we considered other proteins known to be involved in Rad51 removal, including Rrp1, Fhb1, and Srs2 (Huselid & Bunting, 2020). Rrp1 is another member of the Swi2/Snf2 family of translocases that has been implicated in displacing Rad51 from dsDNA, while Fbh1 has been shown to displace Rad51 from ssDNA (Tsutsui et al, 2014). The *rrp1 rad54* double mutant demonstrated a level of Rad51 accumulation that was comparable to that observed in the *rad54* single mutant (Fig S2A and B). This suggests that Rrp1 makes little contribution to Rad51 removal in the absence of Rad54. In contrast, the proportion of cells showing detectable Rad51 fibers was significantly diminished in the *fbh1 rad54* double mutant, and Rad51 foci formation was elevated in these cells (Fig S2A and B). Meanwhile, the proportion of cells exhibiting detectable Rad51 fibers was significantly reduced in the *fbh1 rad54* double mutant, while Rad51 foci were elevated in these cells (Fig S2A and B). Given the increased proportion of cells with short Rad51 filaments in the *fbh1* single mutant, this result suggests that Fbh1 contributes to the conversion of Rad51 foci into fibers. Budding yeast Srs2 possesses the ability to remove Rad51 from ssDNA (Krejci et al, 2003; Veaute et al, 2003), and it has been demonstrated in both budding and fission yeast models that *srs2 rad54* cells are inviable (Palladino & Klein, 1992; Maftahi et al, 2002). We attempted to construct a strain in which a novel *rad54* temperature-sensitive allele (*rad54-ts*, generated for later experiments; see below) was expressed in *srs2* mutant cells to assess the impact of these two mutations on Rad51 accumulation. However, these cells did not grow, even at the permissive temperature (Fig S3), which was unexpected given the very mild phenotype of the *rad54-ts* strain at this temperature (26°C, its characterization is described later). This finding further underscores the intimate interplay between these two genes, thereby pointing to the significant functional overlap between Srs2 and Rad54.

We next sought to determine whether overexpression of Srs2, Fbh1, or Rrp1 is able to alleviate the *rad54* growth defect by integrating the encoding genes into the *ade6*[+] locus under the

control of the tetracycline-inducible promoter (TET-promoter) (Lyu et al, 2024). As a positive control, the Rad54 coding sequence was also placed under the TET-promoter (TET-*rad54*). A *rad54* mutant expressing TET-*rad54* exhibited negligible Rad51 accumulation, regardless of tetracycline supplementation (Fig S4A and B). These results suggest that, even in the absence of tetracycline, the TET-promoter permits low-level expression of *rad54* sufficient to restore function in the *rad54* deletion mutant. In contrast to the TET-*rad54* transgene, each of TET-*rrp1*, TET-*fbh1*, and TET-*srs2* exhibited substantial Rad51 accumulation (comparable to the level of *rad54Δ*) in the absence of tetracycline (Fig S4A and B). Remarkably, when the expression of these proteins was induced, Rad51 accumulation significantly decreased in all strains, with Srs2 induction having the strongest effect, followed by Fbh1 and Rrp1. Together, these data suggest a potential functional overlap between these proteins and Rad54.

### Combination of HR and translesion synthesis mutations causes synthetic growth defects

HR and the translesion synthesis (TLS) pathway have been shown to play redundant roles in the process of bypassing damaged DNA that otherwise results in the obstruction of replication forks. Therefore, we sought to investigate the role of TLS in Rad54-independent damage repair. Rev1, a member of the Y-family DNA polymerase, and Rev3, the catalytic subunit of Polζ, are key components of the TLS pathway that causes the majority of spontaneous mutations (Lawrence, 2004). We combined the *rev1* or *rev3* mutation with the *rad54* null mutation to examine the involvement of TLS in Rad54-independent mechanisms. *rad54* cell growth was further impaired when combined with *rev1* or *rev3*, suggesting that the TLS pathway allows growth when Rad54 is absent (Fig S5A and B). TLS mutations caused similar effects in *rad51* or *rad51 rad54* background stains. These results suggest that the growth reduction observed in these mutants is not unique to the *rad54* mutation, likely due to the synthetic effects brought by the combination of both HR and TLS defects.

### A *rad51-mNeonGreen* transgene allows live cell imaging of Rad51

Next, we sought to understand the mechanism behind Rad51 aggregate formation in the absence of Rad54. To allow for live cell imaging of aggregate formation dynamics, we developed a transgene protein, Rad51-mNeonGreen (mNG), comprised of Rad51 tagged with mNG at its C-terminus and expressed under the control of the *rad51* promoter at the *his3*[+] locus (Fig 4A) (Akamatsu et al, 2007). Cells expressing Rad51-mNG along with native Rad51 showed normal resistance to both bleomycin and HU (Fig S6A and B). Cells expressing Rad51-mNG without WT Rad51 were highly sensitive to both drugs, suggesting that Rad51-mNG by itself is barely functional (Fig S6C). In *Arabidopsis thaliana* and budding yeast, C-terminal modifications with GFP produce a partially functional protein that can carry out meiotic functions but not mitotic ones: these variants bind DNA but fail to strand exchange (Da Ines et al, 2013; Waterman et al, 2019). WT cells expressing Rad51-mNG exhibited similar rates of Rad51 focus/fiber formation

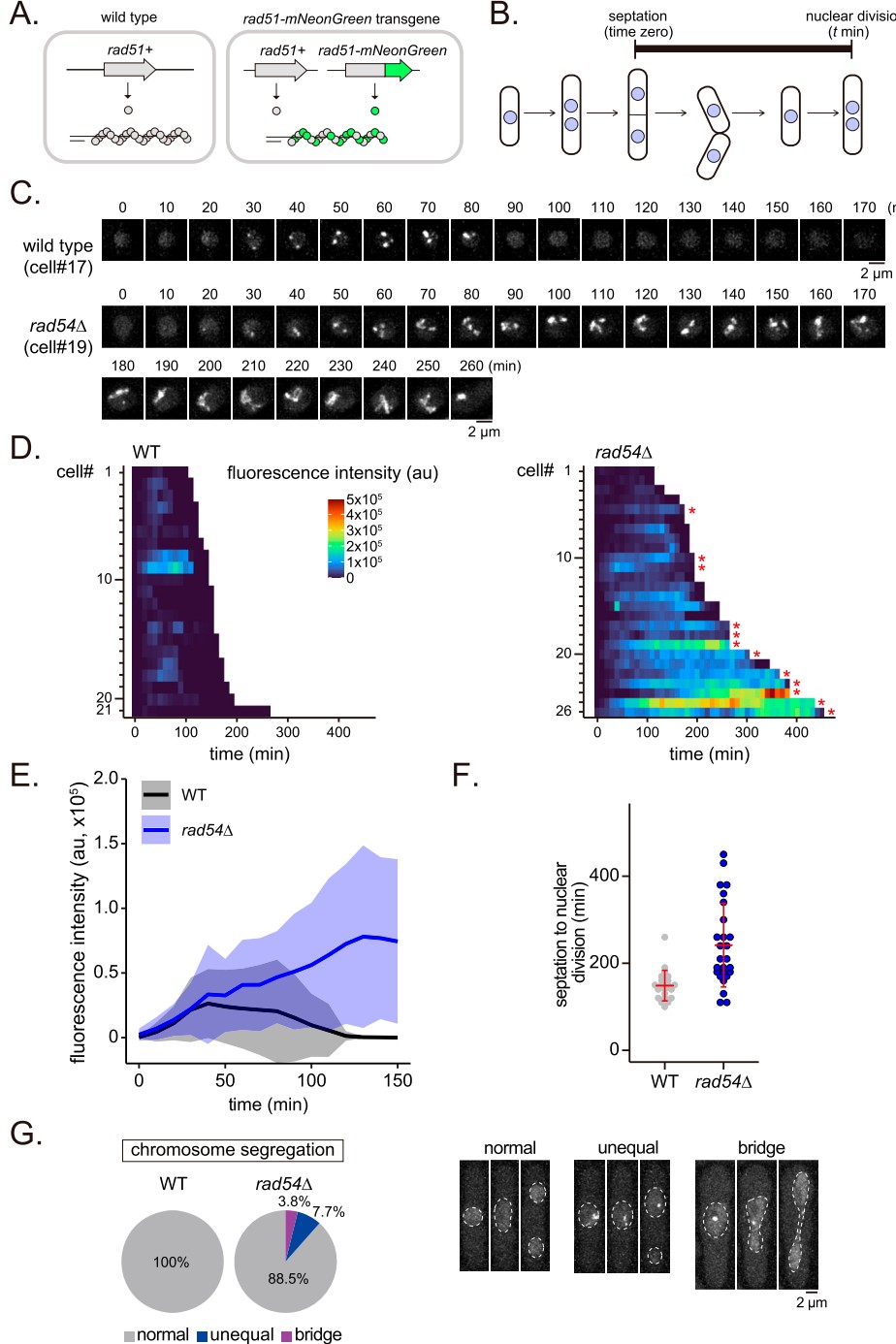

**Figure 4. Live-cell imaging of Rad51 in proliferating cells.**
**(A)** Overview of the *rad51-mNG* transgene system for live-cell imaging of Rad51. A transgene encoding Rad51-mNG was integrated at the *his3⁺* locus. This strain expresses both Rad51 and Rad51-mNG. **(B)** Schematic of the fission yeast cell cycle with starting and ending points of live cell analysis indicated. **(C)** Typical Rad51 localization dynamics in WT and *rad54* mutant cell nuclei. Cell ID (cell#) relates to data shown in (D). **(D)** Quantitative analysis of Rad51 signals per cell from septation to nuclear division. Asterisks indicate cells that underwent nuclear division in the presence of Rad51 bodies. au, arbitrary unit. **(E)** Kinetics of Rad51 appearance in the time course shown in (D). Solid lines indicate the average of replicates; shaded areas represent the SD. The area where the shaded regions (light blue and gray) overlap appears in a blended color, which reflects the overlap of SD between the two strains. **(F)** Duration from septation to nuclear division in the time course shown in (D). Data shown as mean ± SD. **(G)** Ratios for the types of abnormal chromosome segregation detected in (D). Typical nuclear morphologies of *rad54* mutant cells undergoing nuclear division are shown on the right. Strains used: WT (GT506) and *rad54Δ* (GT507).
Source data are available for this figure.

(Fig S7A) and near-identical localization to immuno-stained Rad51 in the *rad54* mutant (Fig S7B). In addition, the kinetics of Rad51 accumulation in bleomycin-treated WT cells were comparable for immuno-stained and Rad51-mNG expressing cells (Fig S7C) and Rad51-mNG signal co-localized with immuno-stained Rad51 (Fig S7D). Expression of this transgene in cells expressing WT Rad51 therefore had no apparent effect on Rad51 function.

We also tagged Rad52 with mCherry to concurrently monitor Rad52 behavior. Rad52-mCherry-expressing cells were resistant to bleomycin and HU, indicating Rad52-mCherry function (Fig S8A and B). Meanwhile, expression of Rad52-mCherry together with ectopic Rad51-mNG resulted in a minor synthetic effect only at high (8 mM) HU concentrations (Fig S8C). Together, these results indicate that in addition to Rad52-mCherry, Rad51-mNG is an invaluable tool for the study of DNA damage in *S. pombe*.

### Timing of Rad51 foci formation is similar in WT and *rad54* mutant cells

We used this Rad51-mNG expressing strain to investigate Rad51 behavior in proliferating cells, from septation to nuclear division (Fig 4B). In WT cells, Rad51-mNG foci transiently appeared ~30 min after septum formation and disappeared before nuclear division (Fig 4C and D, Video 1). The average maximal number of Rad51 foci per cell cycle was 1.8 ± 1.0 (n = 21). In *rad54* mutant cells, the majority of asynchronously growing cells exhibited high levels of Rad51 accumulation, with many showing very slow rates of growth or an inability to divide at all. In order to compare the timing of Rad51 aggregate formation to WT cells, we analyzed a subset of *rad54* mutant cells that underwent nuclear division without aggregated Rad51 (Fig 4C and D, Video 2). The timing of Rad51 foci formation in these cells was nearly identical to that of their WT counterparts (Fig 4E), suggesting that the mechanism underlying Rad51 foci formation is consistent in both strains. However, foci persisted for longer in *rad54* cells (Fig 4D right), with the fluorescence intensity of foci increasing over time (Fig 4E). These analyses revealed that the period between septation and nuclear division was significantly longer in *rad54* mutant cells (Fig 4F; WT mean = 148.6 min, *rad54Δ* mean = 241.5 min).

We observed that a substantial fraction of cells undergoing nuclear division exhibited Rad51 accumulation. In these cells, an accumulated Rad51 mass, termed a Rad51 body, was passed to daughter cells, indicating the intergenerational persistence of Rad51 accumulation (Fig 4D, right; 12 out of 26 cells, marked with an asterisk). Despite the presence of Rad51, the majority of *rad54* cells undergoing nuclear division produced evenly sized daughter nuclei. However, a smaller fraction displayed aberrant phenotypes, including unequal chromosome segregation (7.7%) and chromosomal bridge formation (~3.8%; Fig 4G). These findings suggest that accumulated Rad51 and the associated DNA damage may interfere with the chromosome segregation machinery and/or facilitate interchromosomal associations. However, the relatively low occurrence of aberrant segregation events (~10%) does not account for the high lethality of the *rad54* mutant (~75%).

We were further able to detect a Rad52-mCherry signal in a minority of cells (WT = 19%, *rad54Δ* = 31%; Fig S9A–D). These Rad52 foci invariably co-localized with Rad51. It is known that Rad52 plays a crucial role in Rad51 recruitment to DNA damage sites (Lisby et al, 2004). In fact, we found that the chromosomal localization of Rad51 was entirely dependent on Rad52 (Fig 1C and D). We speculate that the inability to detect Rad52 signal in most cells is due to the low number of Rad52 molecules required for Rad51 focus formation. The appearance of colocalizing Rad52 after the appearance of Rad51 foci suggests that some Rad52 may be involved at a later stage of HR rather than the earlier, DNA damage-related stage (see the Discussion section).

### Intergenerational accumulation of Rad51 leads to hyperaggregation and cell cycle arrest

The deletion of *rad54⁺* has dramatic physiological implications for cells, and we were only able to analyze the small subset of the *rad54Δ* population characterized by phenotypes similar to WT cells. To allow for a more holistic analysis, we isolated a temperature-sensitive allele of *rad54* (*rad54-ts*, see the Materials and Methods section), enabling us to conditionally perturb the function of Rad54 immediately before assessing cellular phenotypes. At the permissive temperature (26°C), *rad54-ts* strains with or without Rad51-mNG showed resistance to both bleomycin and HU, although resistance was not as robust as WT cells (Fig S10A). The cell viability of these strains was comparable to that of WT (Fig S10B). In contrast, at the restrictive temperature (33°C), the level of drug sensitivity was equivalent to the *rad54* null mutant (Fig S10A). In line with this, the viability of these strains at elevated temperature was reduced to the level of the *rad54* null mutant (Fig S10B). We conclude that this *rad54-ts* strain allows for conditional disruption of Rad54 function when cells are subjected to the restrictive temperature (33°C).

The *rad54-ts* allele provides an ideal platform to examine if damage caused by conditional inhibition of Rad54 function is reversed by delayed restoration of Rad54 activity. At permissive temperature, 92.5% of cells were viable, whereas viability dropped to 27.5% in cells maintained under restrictive conditions (Fig S10C). Cells transferred to permissive temperatures after 24 h incubation under restrictive conditions exhibited low viability (36.7%) comparable to cells kept at restrictive temperature for the entire duration of the experiment. This suggests that cells accrue irreversible damage, such as chromosome missegregation, in the absence of Rad54.

We next examined the behavior of Rad51 when *rad54-ts* cells were shifted from permissive to restrictive temperatures (Fig 5). Initially, we considered cells without Rad51 bodies at time zero and observed these cells from septation to nuclear division (the experiment concluded at 420 min if the cell did not enter mitosis). Rad51 behavior in these cells closely mirrors that observed in *rad54* null mutant cells. In comparison with WT cells incubated at 33°C, *rad54-ts* cells accumulated Rad51 (Fig 5A, left and center panels) and exhibited a delay in the onset of nuclear division, with division requiring on average 225 min in *rad54-ts* cells versus 156 min in WT cells (Fig 5B). Of 30 examined cells, one failed to undergo mitosis within 420 min; instead, a high-intensity Rad51 signal accumulated in the nucleus of this cell (Fig 5A center panel, cell #30). Remarkably, a Rad51 signal was evident in a majority of cells that entered mitosis (58.6%), with subsequent transmission of Rad51 bodies to daughter cells resulting in nuclei carrying a Rad51 body before entering S phase in daughter cells.

We next examined such cells carrying Rad51 in the nucleus upon septation (Fig 5A, right panel). We observed that nuclear Rad51 fluorescence intensity continued to increase significantly throughout the experiment, suggesting sustained and cumulative accumulation. 50% of these Rad51-positive cells entered mitosis, indicating a much lower rate of successful cell cycle completion compared with those starting without Rad51 bodies (96.7%). Of initially Rad51-positive cells that were able to begin mitosis, 86.4% underwent nuclear division in the presence of Rad51 bodies. When initially Rad51-positive cells were unable to undergo mitosis, we observed extremely high levels of Rad51 in many cells, indicating potential permanent arrest of the cell cycle

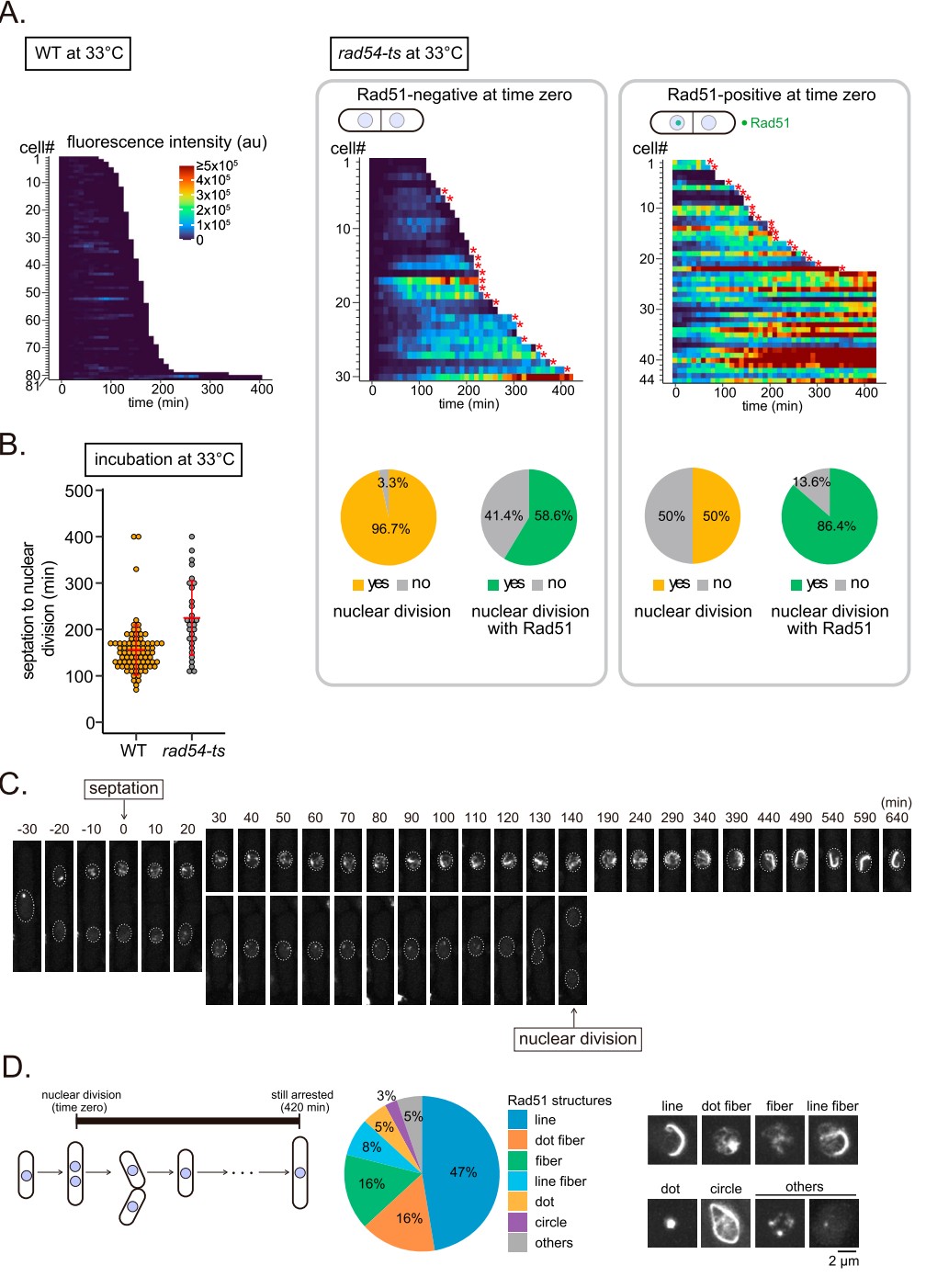

**Figure 5. Inherited Rad51 bodies often develop into Rad51 hyper-aggregates and result in cell cycle extension.** **(A)** Quantitative analysis of Rad51 signals per cell from septation to nuclear division. Asterisks indicate cells that underwent nuclear division in the presence of Rad51 bodies. au, arbitrary unit. Saturated intensities (>5 × 10⁵ au) appear in middle and right panels (see also Fig S11). **(B)** Duration from septation to nuclear division for indicated strains. Data shown as mean ± SD. For *rad54-ts*, only cells lacking Rad51 bodies upon septation that underwent nuclear division were measured (WT, n = 81; *rad54-ts*, n = 29). **(C)** An example of a *rad54-ts* cell that entered mitosis in the presence of a Rad51 body. A daughter cell that inherited a Rad51 body remained undivided at 640 min with a Rad51 hyper-aggregate. **(D)** Schematic of the fission yeast cell cycle with definition of "arrest" indicated (420 min). Rad51 aggregate morphologies in cells that did not divide were categorized as indicated (n = 38). Strains used: WT (GT295) and *rad54-ts* (GT696). WT, wild type. Source data are available for this figure.

(Fig 5A, right panel; Fig S11). Rad51 bodies transmitted during mitosis appeared to persist and develop into extensive Rad51 aggregates (Fig 5C); these inherited Rad51 aggregates exhibited distinct morphological characteristics, including lines, dots, and fibers, with lines being observed in 50% of arrested cells (Fig 5C and D). In conclusion, we infer that the extensive accumulation of Rad51 aggregates observed in the majority of the *rad54* mutant cell population is the result of the transmission of DNA-associated Rad51 over cell generations.

## Rad51 foci appear at late S phase

Due to the role of Rad51 in DNA repair and the strong effect of Rad51 bodies on cell cycle progression, we sought to determine the cell cycle stage at which Rad51 foci appear. To determine this, we constructed a strain expressing a PCNA-RFP fusion protein (mCherry-Pcn1) as a marker of DNA replication (Meister et al, 2007; Vještica et al, 2020). Expression of PCNA-RFP resulted in a similar level of resistance to DNA damage and replication stress

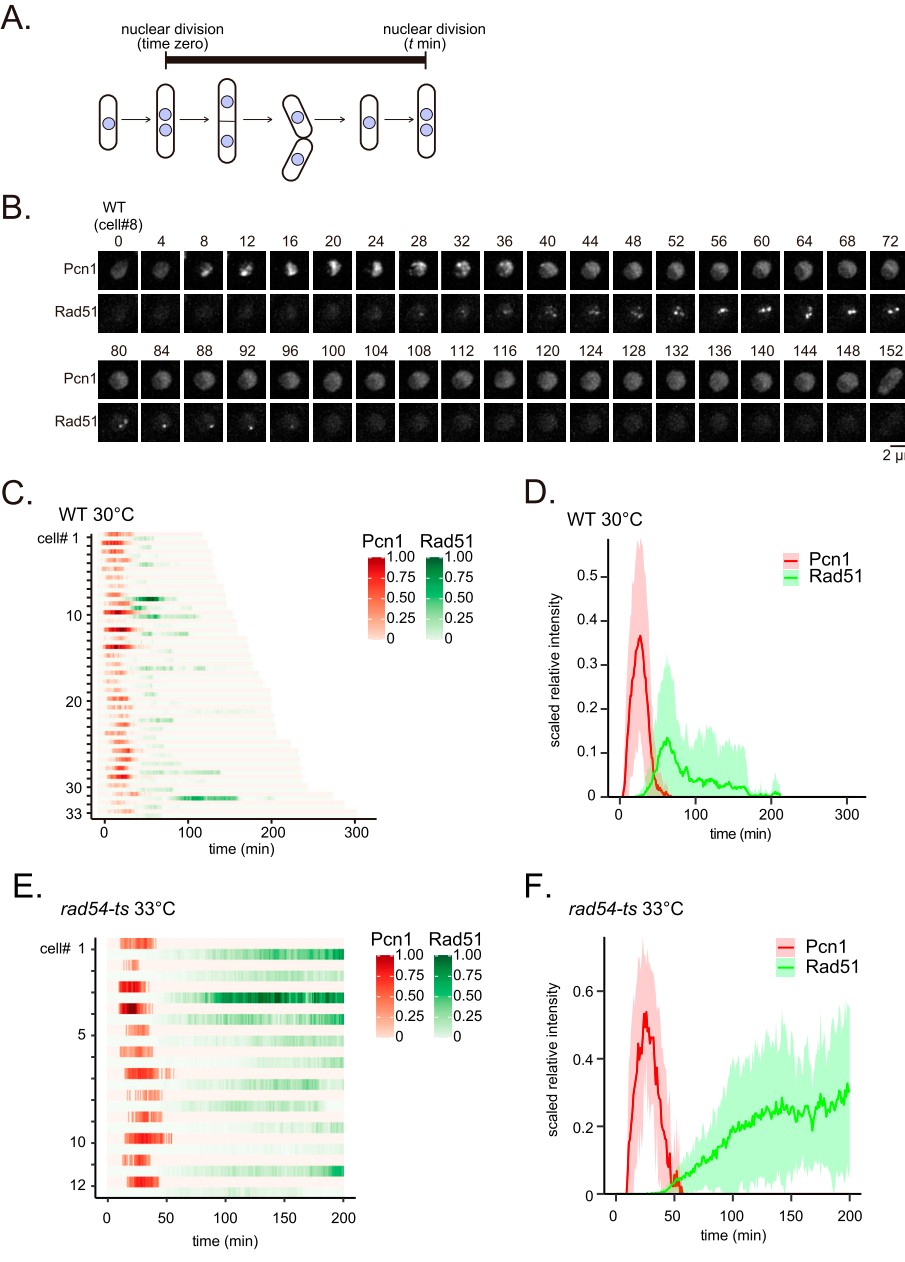

**Figure 6. Rad51 foci appear at the late stage of S phase.**
**(A)** Schematic of the fission yeast cell cycle with start and end points of live cell analysis indicated. **(B)** Representative morphologies of Pcn1 (PCNA) and Rad51 between nuclear division events in WT cells expressing mCherry-Pcn1 and Rad51-mNG. **(C)** Quantitative analysis of Pcn1 and Rad51 between nuclear division events. Maximum intensities were normalized to 1.0 for each cell. **(D)** Results shown in (C) were plotted as a line graph. Solid lines represent the average of 33 determinations, and shaded areas SD. **(E)** Quantitative analysis of Pcn1 and Rad51 signals in individual *rad54-ts* mutant cells between nuclear division events at 33°C. **(F)** Line graph of data shown in (E). Strains used: WT (GT991), *rad54-ts* (GT1207). WT, wild type.
Source data are available for this figure.

as observed in WT cells, suggesting that PCNA-RFP is functional in HR (Fig S12A). Defining nuclear division events as time zero, we found that PCNA signals begin to increase within 10 min and subsequently return to background levels by 60 min, indicating the completion of the majority of genomic DNA replication (Fig 6A–C). The appearance of Rad51 foci coincided with the timing of PCNA signal decrease, which is consistent with late S phase (Meister et al, 2007). Meanwhile, the timing of PCNA appearance and disappearance events was remarkably reproducible, showing little variation among the cells examined (Fig 6C). While the timing of Rad51 appearance was also consistent, significant variation was observed in the kinetics of disappearance,

suggesting that the dissolution of Rad51 foci in particular is stochastic in nature (Fig 6D). The appearance of Pcn1 and Rad51 in *rad54-ts* mutant cells grown at the restrictive temperature was comparable to that in WT cells grown at 30°C (Fig 6E and F), as well as to WT cells grown at the restrictive temperature (Fig S12B and C). These data indicate that Rad51 accumulation begins during late S phase in both WT and *rad54* mutant strains. We also confirmed the timing of Rad51 disappearance using WT cells expressing both Rad51-mNG and Sad1-mCherry, a spindle pole body (SPB) component that can be used to ascertain the onset of M phase (i.e., mitotic entry) (Nakazawa et al, 2008). These strains also exhibited a similar resistance to DNA damage and

replication stress compared with WT cells, confirming the integrity of their HR machinery (Fig S12A). Rad51 signals disappeared well before the duplication of Sad1 signals (about 65 min earlier on average), suggesting that Rad51 dissolution precedes mitotic entry (Fig S12D–F).

### Formation of Rad51 bodies depends on the long-range resection machinery

The temporary appearance of Rad51 foci at late S stage, observed in both WT cells and the *rad54* mutant, raises the possibility that Rad51 accumulation and the subsequent aggregation caused by the loss of Rad54 are triggered by some form of DNA damage associated with replication. Rad51 accumulation in the *rad54* mutant requires Rad52, Rad57, and Sfr1 (Fig 1C and D), auxiliary factors that are also involved in the localization of Rad51 to DSB (Akamatsu et al, 2007; Lorenz et al, 2009; Tsubouchi et al, 2021). Loading of Rad51 onto DNA at sites of damage typically occurs where ssDNA is exposed by DNA nucleases. The resection of DSB ends is thought to be initiated by the collaborative action of the MRN complex and Ctp1, which allows the entry of the long-range resection machinery consisting of Exo1, and Rqh1 and Dna2 (Cejka & Symington, 2021). The absence of both Exo1 and Rqh1 abolishes the majority of resection activities, leaving short 3′-overhangs created by MRN-Ctp1 at DSB ends (Yan et al, 2019). We reasoned that if the formation of Rad51 accumulation observed in the *rad54* mutant is triggered by DNA breaks, it would require end resection activities.

Either or both of the *exo1* and *rqh1* mutations were introduced into the *rad54* mutant and the implications for Rad51 accumulation were determined by fluorescence microscopy (Fig 7A and B). Both *rad54* or *rad54 rqh1* double mutant cells showed similar proportions of Rad51 foci and fibers. In contrast, the absence of Exo1 caused a stark reduction in long fibers and further reduced the cell growth of the *rad54* mutant (Fig S13). The loss of both Exo1 and Rqh1 dramatically reduced the amount of all classes of Rad51 accumulation.

We next examined the dynamics of Rad51 localization using confocal microscopy. The robust accumulation of Rad51-mNG normally observed in *rad54* cells was greatly reduced in the absence of both Exo1 and Rqh1 (Fig 7C–E). Interestingly, the time between septation and nuclear division was comparable (>220 min) for *rad54* cells and the *rad54 exo1 rqh1* mutant, suggesting that DNA damage is detected similarly in these strains (Fig 7D and F).

To further understand the relationship between ssDNA and Rad51 accumulation, we next observed strains expressing tagged Ssb1 (the largest subunit of RPA, a eukaryotic ssDNA-binding protein) and Rad51. Expression of Ssb1-mCherry did not affect resistance to DNA damage and replication stress (Fig S14A and B). Rad51 foci showed strong colocalization with RPA in both WT and *rad54* mutant strains (75% and 85.3%, respectively) (Fig 8A and B). When Rad51 showed both foci and fiber formation in *rad54* mutant cells, RPA mostly associated with foci only (67.7%), while 26.5% of cells exhibited colocalization with both foci and fibers. These

results strongly suggest that Rad51 accumulation is closely associated with ssDNA.

If Rad51 aggregation in *rad54* mutant cells is triggered by the production of ssDNA, similar results should be recapitulated upon the artificial induction of DNA breaks. Initially, we treated WT cells with bleomycin and HU, which induce DNA breaks and replication stress, respectively. Cells treated with bleomycin exhibited a robust accumulation of Rad51 while HU treatment caused only a marginal accumulation (Fig S15A–C). Treatment of both *rad54* and *rad54 exo1 rqh1* triple mutant cells with bleomycin resulted in marginal induction in *rad54 exo1 rqh1* triple mutant cells compared with the *rad54* parental strain, in which accumulation was robust in most cells (Fig S16A–C).

Taken together, these data strongly suggest that ssDNA exposed by the long-range resection machinery initiates the formation of Rad51 accumulation in the *rad54* mutant. Thus, Rad54 promotes the repair of spontaneous late-S DNA breaks, likely including SSBs as well as DSBs, that would otherwise result in intergenerational Rad51 aggregation.

## Discussion

Using fission yeast as a model, we observed robust accumulation of Rad51 in vegetatively growing *rad54* mutant cells. The accumulation of Rad51 was seen both by immunostaining and by the use of a recessive Rad51-mNG protein. By developing a live-cell imaging protocol to track Rad51, we identified the origin and fate of Rad51 aggregates formed in these mutants. Our findings strongly suggest that DNA breaks arising in late S phase act as the initiators of subsequent Rad51 accumulation. In both WT and *rad54* mutant cells, Rad51 initially appears as foci during late S phase. However, while these foci rapidly resolve in WT cells, they continue to grow during the prolonged G2 phase of *rad54* mutants. The accumulated Rad51 often persists into M phase, forming a Rad51 body that is distributed into daughter cells alongside chromosomes. Persistent growth of this Rad51 mass eventually leads to the formation of large Rad51 aggregates frequently associated with cell cycle arrest. Thus, in fission yeast, the primary role of Rad54 in vegetative cells appears to be facilitating the repair of DNA breaks arising in late S phase, preventing the runaway intergenerational aggregation of Rad51.

### Rad51 aggregate formation in the *rad54* mutant is triggered by DNA breaks

Our results indicate that the robust accumulation of Rad51 caused by loss of Rad54 is triggered by DNA break-initiated ssDNA production. First, Rad51 accumulation requires the presence of Rad52, Rad57, and Sfr1—major auxiliary factors of Rad51 that are also essential for Rad51 recruitment to DSBs. Second, Rad51 accumulation in the *rad54* mutant is initiated by Rad51 foci periodically formed at late S phase in WT cells. It is therefore likely that Rad51 foci are associated with replication-related DNA damage. Furthermore, hyperaccumulation of Rad51 is largely

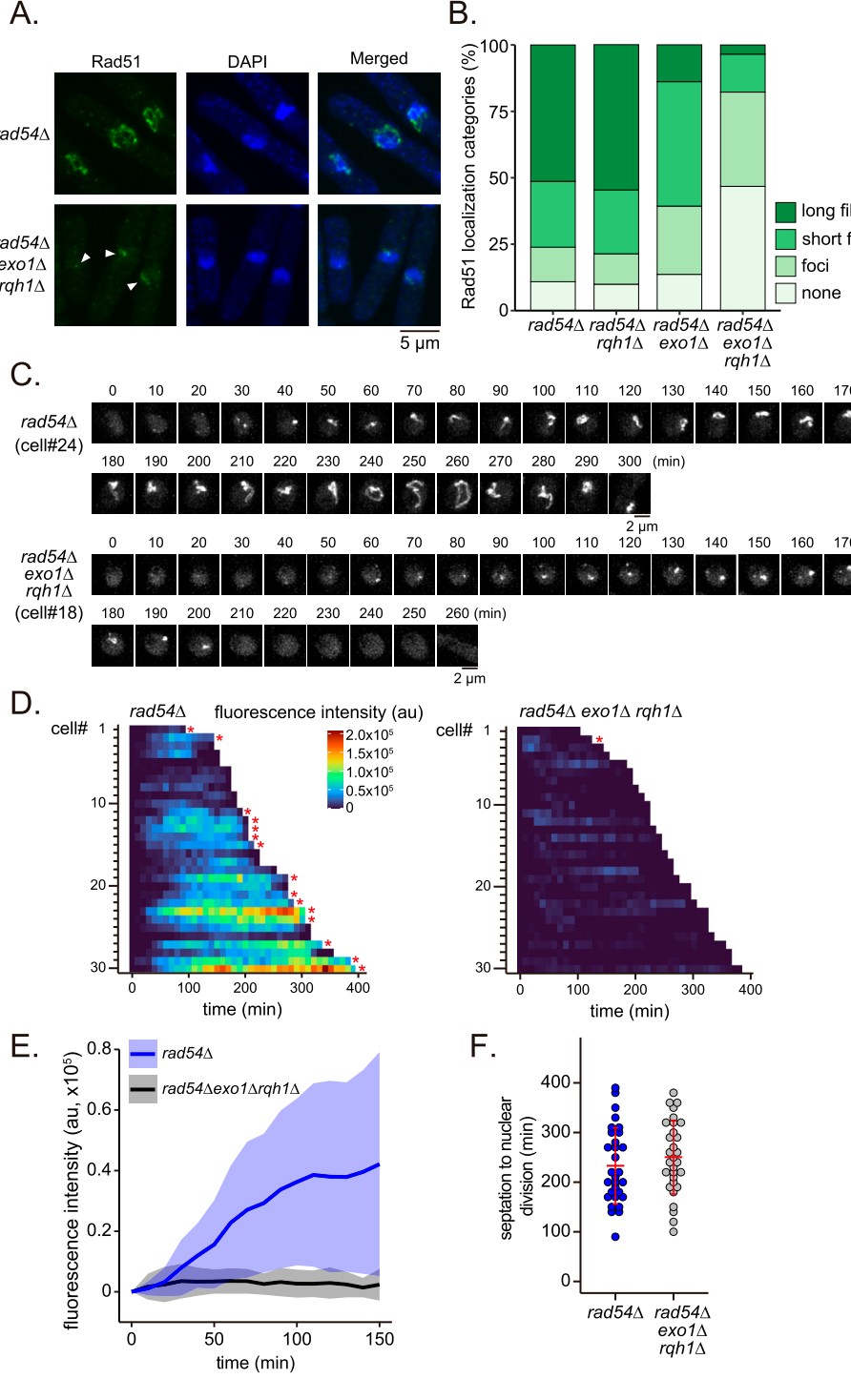

**Figure 7. Rad51 accumulation in *rad54* mutant cells depends on long-range resection machinery.**

**(A)** Representative Rad51 morphologies in the indicated strains. Arrowheads indicate the location of Rad51. **(B)** Categorization of Rad51 morphologies. Structures were classified as shown in Fig 1. n: *rad54Δ* = 754; *rad54Δ rqh1Δ* = 733; *rad54Δ exo1Δ* = 677; *rad54Δ exo1Δ rqh1Δ* = 665. Strains used: *rad54Δ* (GT422), *rad54Δ exo1Δ* (GT1175), *rad54Δ rqh1Δ* (GT1180), and *rad54Δ exo1Δ rqh1Δ* (GT1171). **(C)** Typical Rad51 localization dynamics in nuclei of the indicated strains. Cell ID (cell#) relates to data shown in (D). **(D)** Quantitative analysis of Rad51 signals per cell from septation to nuclear division. Asterisks indicate cells that underwent nuclear division in the presence of Rad51 bodies. au, arbitrary unit. **(E)** Kinetics of Rad51 appearance in the time course shown in (D). Solid line, average of replicates; shaded area, SD (n: *rad54Δ* = 30; *rad54Δ exo1Δ rqh1Δ* = 30). **(F)** Duration from septation to nuclear division in the time course shown in (D). Data shown as mean ± SD. Strains used: *rad54Δ rad51-mNG* (GT325), *rad54Δ exo1Δ rqh1Δ rad51-mNG* (GT1134).
Source data are available for this figure.

dependent on the long-range resection machinery, whose main function is to create a long 3′-ended ssDNA overhang that eventually serves as a landing pad for Rad51. Recently, the dynamics of Rad51 nucleoprotein filaments following DSB formation were observed in live budding yeast cells (Liu et al, 2023a). Some of these structures are reminiscent of those seen in the fission yeast *rad54* mutant at later time points. This similarity provides further support for the association of the accumulated Rad51 in the fission yeast *rad54* mutant with DNA breaks.

Our data also indicate that the DNA damage checkpoint surveillance mechanism is constitutively active in *rad54* mutant cells. In the absence of Rad54, Chk1, which is mainly responsible for monitoring DNA damage, exhibited a clear gel mobility shift that is consistent with a phosphorylated, active state. In contrast, Cds1,

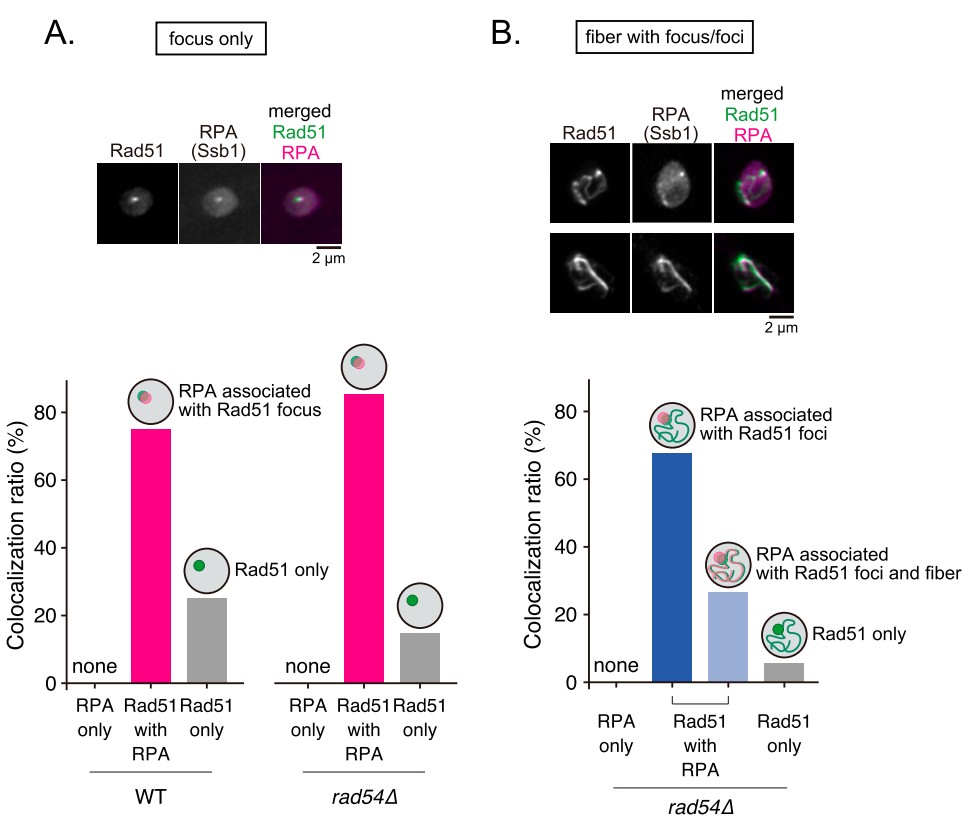

**Figure 8. Rad51 aggregates are often associated with RPA.**
Representative images and quantitative analyses of cells expressing both Rad51-mNG and RPA (Ssb1)-mCherry. **(A)** Nuclei with a focus-type localization were examined for the association of Rad51 and RPA. n: WT = 292; *rad54Δ* = 190. **(B)** Nuclei exhibiting both fiber and focus/foci type localizations were examined for the association of Rad51 and RPA. (n = 468). Cells in which RPA signal overlapped exclusively with fiber-type Rad51 were not observed. Strains used: *rad51-mNG ssb1-mCherry* (GT573), *rad51-mNG ssb1-mCherry rad54Δ* (GT435). WT, wild type. Source data are available for this figure.

which is mainly charged with replication stress during S phase, was not associated with such a shift. Indeed, the marked increase in cell length, which is a hallmark of *rad54* mutant cells, was not observed in the absence of Chk1. These observations provide further support for our contention that DNA breaks act as a trigger for the formation of Rad51 aggregates.

Our findings revealed that Rad54 was not indispensable for the repair of DNA breaks that occur during late S phase. Approximately 40% of *rad54* mutant cells demonstrated the capacity to successfully resolve Rad51 foci and proceed to mitosis (Fig 4D right and Fig 5D middle). DSB repair is heavily dependent on HR in fission yeast. Indeed, *rad54* is essential in homothallic strains in which DSB-initiated mating type switching is active (Roseaulin et al, 2008). Therefore, some of the Rad51 foci that are prominent in the *rad54* mutant may represent SSB-associated ssDNA.

In budding yeast and human cells, it has been proposed that an equilibrium exists between the spontaneous binding of Rad51 to undamaged dsDNA and its dissociation mediated by Rad54-family translocases (Holzen et al, 2006; Shah et al, 2010; Mason et al, 2015). Budding yeast *rdh54* and *rad54 rdh54* mutants exhibit significantly higher levels of spontaneous Rad51 foci compared with the WT strain. Limited colocalization of Rad51 with RPA under these conditions suggests that these Rad51 foci are not associated with DNA damage. Notably, the *rdh54 rad54 uls1* triple mutant displays an even greater

accumulation of Rad51 foci than any combination of these mutations alone. In contrast, somatic accumulation of Rad51 in the fission yeast *rad54* mutant appears to be associated with DNA breaks, as discussed above. Because Rdh54 is meiosis-specific in fission yeast (Catlett & Forsburg, 2003), *rad54* deletion in this organism should be phenotypically equivalent to the *rad54Δ rdh54Δ* mutant in budding yeast. This suggests that the nature of Rad51 accumulation when Rad54-family translocases are compromised is markedly different between budding and fission yeast. This apparent discrepancy may reflect differences in the frequency of DNA breaks spontaneously arising during normal proliferation. In fission yeast, ~2 Rad51 foci emerge per cell cycle. Given that both yeast species rely heavily on HR for DSB repair, a higher frequency of spontaneous DNA breaks in fission yeast could result in greater Rad51 accumulation. Therefore, non-damage-associated Rad51 likely constitutes only a small fraction of the robust Rad51 accumulation observed in this organism.

## Origins of DNA breaks during late S phase

The kinetics of Rad51 foci appearance are comparable in WT and *rad54* mutant cells, indicating that similar events during late S stage trigger foci formation in both strains. This raises the question of what triggers DNA break formation specifically at late S stage. We speculate that such break formation is related to the progression

of the replication fork. Impairment of replication fork progression leads to replication stress and a stalled replication fork (Zeman & Cimprich, 2014). One typical chromosomal feature that interferes with fork progression is heterochromatin. In mammalian cells, DNA replicated in late S phase is associated with transcriptionally inactive heterochromatin (O'Keefe et al, 1992). In fission yeast, subtelomeric heterochromatin is replicated at late S phase, although centromeric and the silent mating type locus heterochromatin replicate during early S phase (Kim et al, 2003; Hayashi et al, 2009). It is crucial to identify what triggers the late S-arising DNA breaks.

Stalled replication forks are associated with both SSBs and DSBs. Recent studies have highlighted the critical role of Rad51 binding to dsDNA in protecting stalled replication forks (Halder et al, 2022; Liu et al, 2023b). In human cells, G2-specific phosphorylation of Rad54 by the Nek2 kinase ensures its activation during this phase of the cell cycle (Spies et al, 2016). The absence of Rad54 activation during S phase stabilizes Rad51 filaments, thereby contributing to replication fork stability. When Rad54 is deficient, accumulated Rad51 at replication forks may serve as a nucleation site, promoting further expansion of Rad51 filaments.

### Rad54 function in vegetatively growing cells

Rad51 foci in WT cells exhibit a striking periodicity, appearing transiently during late S phase in each cell cycle. As discussed above, these foci are likely triggered by DNA breaks. Given that massive Rad51 aggregates form in the absence of Rad54, the primary role of Rad54 in normally proliferating cells appears to be facilitating the repair of late S-arising DNA breaks through Rad51-mediated HR.

The post-DSB role of Rad54 in HR is well-documented (Ceballos & Heyer, 2011). In budding yeast, Rad51 can localize to a DSB site even in the absence of Rad54, indicating that Rad54 functions downstream of Rad51 presynaptic filament formation (Sugawara et al, 2003; Lisby et al, 2004; Renkawitz et al, 2013; Liu et al, 2023a). Mechanistically, Rad54 facilitates Rad51-mediated homology search and strand invasion (Petukhova et al, 1998; Mazin et al, 2000; Van Komen et al, 2000; Sigurdsson et al, 2002; Crickard et al, 2020). Rad54 is also involved in dissociating Rad51 from heteroduplex DNA (Bugreev et al, 2007). Our observations in fission yeast are broadly consistent with these known roles of Rad54 at the synapsis and post-synapsis stages of HR, which explains the Rad51 accumulation observed in rad54 mutant cells.

Given that Rad54 functions primarily during the synaptic phase and thereafter, Rad51 aggregates likely represent Rad51 molecules accumulating on gradually exposed ssDNA. This view is consistent with data showing that Rad51 aggregation is suppressed when DNA end resection is compromised. However, it is possible that exposed ssDNA is only required to initiate Rad51 aggregation, and that Rad51 filament formation may extend beyond the ssDNA/dsDNA junction, covering dsDNA to form larger aggregates. Indeed, Rad54 has been shown to remove Rad51 from dsDNA, but not from ssDNA (Solinger et al, 2002; Ceballos & Heyer, 2011). We observed that only one-third of Rad51 filaments co-localize with RPA,

possibly reflecting association of Rad51 with dsDNA. However, this observation might simply reflect the mutually exclusive binding of Rad51 and RPA to ssDNA, as only one of these ssDNA-binding proteins can bind a given region at a time.

### Mechanisms supporting viability in the absence of Rad54

Although rad54 mutant cells exhibit strong growth defects, about a quarter of these cells remain viable, suggesting that alternative mechanisms allow their survival. We examined a group of proteins involved in Rad51 removal—namely, Srs2, Fbh1, and Rrp1 (Huselid & Bunting, 2020). When overexpressed in rad54 mutant cells, any of these three proteins substantially reduced Rad51 accumulation. This observation suggests that Srs2, Fbh1, and Rrp1 can negatively regulate Rad51 accumulation. *Saccharomyces cerevisiae* Srs2 and *S. pombe* Fbh1 are known to displace Rad51 from ssDNA, while *S. pombe* Rrp1 removes Rad51 from dsDNA (Krejci et al, 2003; Veaute et al, 2003; Tsutsui et al, 2014; Muraszko et al, 2021). Beyond direct physical displacement of Rad51, it is also possible that Fbh1 and Rrp1 reduce Rad51 levels by promoting its degradation. Fbh1 contains an F-box motif and functions with the SCF complex as an E3 ubiquitin ligase targeting Rad51 (Tsutsui et al, 2014). Similarly, Rrp1 contains a RING domain and also serves as an E3 ligase for Rad51 (Muraszko et al, 2021).

Rrp1, however, is unlikely to play a major role in the Rad54-independent repair pathway because the rrp1 rad54 double mutant is essentially indistinguishable from the rad54 single mutant. In contrast, the srs2 null mutation, when combined with rad54, is lethal in both *S. cerevisiae* and *S. pombe*, suggesting a close functional relationship between the two genes (Palladino & Klein, 1992; Maftahi et al, 2002; Zinovyev et al, 2013). This synthetic lethality prevented us from examining the effect of Srs2 on Rad51 accumulation in the rad54 mutant. We therefore attempted to use the rad54-ts mutation, but were unable to isolate a srs2 rad54-ts double mutant even at the permissive temperature. Because the rad54-ts mutant exhibits only a very subtle phenotype under permissive conditions, this finding further supports a strong genetic interaction between srs2 and rad54. Thus, Srs2 likely plays a major role in the Rad54-independent repair pathway. In the fbh1 rad54 double mutant, the proportion of cells exhibiting Rad51 fibers was markedly reduced, while the fraction of cells with Rad51 signals was comparable to that in the rad54 mutant. This pattern resembles the characteristic Rad51 localization observed in fbh1 mutants, where cells with long Rad51 fibers are relatively rare. Given that Rad51 fibers originate from foci, Fbh1 appears to function in converting Rad51 foci into fibers. If the fiber represents a long Rad51 nucleoprotein filament, the requirement for Fbh1 suggests that occasional removal of Rad51 from DNA may be necessary for its continuous extension.

HR and the TLS pathway collaborate to overcome DNA damage that blocks replication forks. Rev1, a Y-family DNA polymerase, and Rev3, the catalytic subunit of Polζ, are major players in the TLS mechanism responsible for most spontaneous mutations (Lawrence, 2004). The already severely impaired growth phenotype of rad54 cells was further exacerbated by introduction of rev1 or rev3 mutations. Similar results were observed when these TLS mutations were introduced into rad51 or rad51 rad54 double

mutants, consistent with the cooperative role of HR and TLS during replication. It is possible that the accumulated Rad51 in *rad54* mutants interferes with TLS activity. However, we found it difficult to draw definitive conclusions about growth differences between strains such as *rev1 rad51* and *rev1 rad54*. This difficulty arises primarily because introducing *rev1* or *rev3* into either the *rad51* or *rad54* mutant background results in extremely impaired growth phenotypes. Furthermore, these double and triple mutants often exhibited notable colony size variation, likely due to increased genome instability, further complicating direct comparisons.

### Rad51 accumulation and the cell cycle progression

Live-cell imaging of Rad51 revealed a close correlation between the intensity of Rad51 signals and the duration of the G2 phase. Given that Rad51 accumulation is triggered by DNA breaks and Rad51 is recruited to damaged DNA, it is likely that the DNA damage checkpoint is activated in a subset of cells attempting to repair such damage. This idea is supported by the sustained activation of Chk1 kinase in the *rad54* mutant, where Rad51 accumulation is prominent (Fig 3B). In the absence of Chk1, cell length is reduced to that of WT cells. Similarly, Rad51 levels are much lower than in the *rad54* single mutant, although still higher than in WT. These observations indicate that DNA damage, accompanied by Rad51 recruitment, causes cell cycle delay via DNA damage checkpoint activation, and that Rad51 accumulation occurs gradually in cells characterized by slowed cell cycle progression.

As Rad51 accumulation inhibits cell growth, the reduced Rad51 accumulation observed in the *rad54 chk1* double mutant cells might be expected to allow improved cell growth. However, we observed that colony size is much smaller than in the *rad54* single mutant colonies (Fig S1B), indicating that the *chk1* mutation exacerbates the growth defect caused by the *rad54* mutation. This is likely due to a combination of HR defects and impaired DNA damage checkpoint function, resulting in attempted mitosis with unrepaired DNA. The deleterious effects of checkpoint failure therefore appear to outweigh the modest benefit of reduced Rad51 accumulation.

Similarly, under conditions in which long-range exonuclease activity is compromised (i.e., in the *exo1 rqh1 rad54* triple mutant), Rad51 accumulation is substantially reduced, presumably due to limited resection (Fig 7D). Despite this reduction, the cell cycle remains delayed, indicating persistent checkpoint activation. These observations suggest that short-range resection activity provided by the MRN–Ctp1 complex is sufficient to sustain checkpoint signaling. Furthermore, long-range exonuclease activity may contribute to pathways independent of HR, and its loss could also negatively impact cell growth. We speculate that these negative effects outweigh the relatively minor benefit conferred by reduced Rad51 accumulation.

### Rad51 aggregate formation via intergenerational transmission of Rad51 bodies

The use of a novel *rad54-ts* mutant allele allowed us to examine the dynamics of Rad51 accumulation from the initial nucleation event. Such cells are able to initiate a new cell cycle without a

Rad51 body, but Rad51 accumulation is clearly pronounced in the absence of Rad54 function, with signal observed to increase during passage from S to G2 phases. Importantly, more than 50% of these cells underwent nuclear division in the presence of accumulated Rad51, resulting in daughter cells inheriting a persistent Rad51 body upon septation (entry to G1 phase). These cells tend to accumulate further Rad51 at these structures, with some developing massive Rad51 aggregates that are associated with prolonged G2 phase. We determined that 50% of cells containing such aggregates undergo cell cycle arrest, whereas the remainder continue with mitosis, often with the Rad51 mass persisting and ultimately being inherited by the next generation. Interestingly, observation of Rad51 dynamics revealed that transmitted Rad51 bodies do indeed seed further development of Rad51 accumulation, and that these ultimately form harmful aggregates. We therefore conclude that the striking Rad51 aggregates observed in *rad54* mutant cells are formed through cumulative recruitment of Rad51 and the intergenerational transmission of resulting Rad51 bodies from mother to daughter cells.

We also observed that many mutant cells did enter mitosis even with pronounced Rad51 aggregation. These likely reflect a subset of cells that are able to overcome the DNA damage checkpoint and resume the cell cycle. The DNA damage checkpoint is regulated through a cascading series of protein phosphorylation events that are counteracted by a group of protein phosphatases (Clémenson & Marsolier-Kergoat, 2009). We speculate that the degree of G2 phase extension is directly impacted by the magnitude of DNA damage accumulated in the cell. Accordingly, accumulation of DNA damage to a point that overwhelms the adaptive phosphatases activities implicated in DNA damage response can cause a near-permanent cell cycle arrest.

An enzymatically broken chromosome can be transmitted across generations in both budding and fission yeast. In budding yeast, broken acentric sister chromatids often co-segregate into either the mother or daughter cell and can be repaired after being segregated over several generations (Kaye et al, 2004). In fission yeast, a semi-artificial chromosome with an unrepaired single-ended DSB undergoes cycles of replication and end resection across multiple generations. These cycles persist until the chromosome either forms stable rearrangements or is eventually lost (Pai et al, 2023). The intergenerational accumulation and transmission of Rad51 mirrors the transgenerational transmission of such broken chromosomes. We propose that DNA breaks specifically arising in late S phase act as natural initiators of the intergenerational transmission of broken chromosomes, while their association with Rad51 further exacerbates chromosomal instability.

### Intergenerational transmission of Rad51 and genome instability

The transmission and gradual development of Rad51 aggregates across cell generations have unique implications. As Rad51 accumulations appear to originate in DNA break events, Rad51 bodies likely indicate transmittable DNA damage. If a daughter cell receives not just damaged DNA but also certain associated proteins, the fate of such DNA damage may be constrained by these proteins. The presence of Rad51 or a possible

superstructure comprising DNA and Rad51 might interfere with subsequent DNA replication, leading to the expansion of DNA damage and associated Rad51. In vertebrate cells, it is known that certain DNA damage, originating from under-replicated genomic loci, can be transmitted to a daughter cell, where it is immediately protected by 53BP1 to form a DNA-protein body (Lukas et al, 2011). These 53BP1 nuclear bodies are implicated in shielding vulnerable DNA damage sites against further erosion. In *Caenorhabditis elegans*, DNA damage that is inherited from the male parent leads to the formation of heterochromatin, thereby impeding the action of homologous recombination mechanisms on damaged DNA (Wang et al, 2023). Similarly, the transmission of damaged DNA associated with Rad51 from mother to daughter in *S. pombe* may interfere with the genome maintenance mechanisms that operate normally, thereby triggering genome destabilization.

The intergenerational transmission of Rad51 bodies also provides a unique insight into cancer development. Recent proposals suggest that DNA damage initiates mutational cascades that extend over multiple cell generations, potentially driving the evolution and diversification of cancer genomes (Aitken et al, 2020; Umbreit et al, 2020; Lezaja & Altmeyer, 2021). Cancer development often involves defects in HR and weakened checkpoint surveillance for DNA damage/replication. Under such circumstances, the likelihood of Rad51 associated with damaged DNA being transmitted to a daughter cell would be much higher. The co-transmission of DNA damage and associated proteins to the next cell generation might serve as a driver to accelerate genome instability in some cancers.

# Materials and Methods

All strains and primers used in this study are listed in Tables S1 and S2, respectively.

## Yeast strains

All *S. pombe* strains are isogenic derivatives of GT414 unless otherwise stated. GT414 carries *mat1P-Δ17*, where programmed DSB at the mating type locus does not occur (Arcangioli & Klar, 1991). Gene deletion and tagging were conducted by PCR-mediated homology-based gene targeting (Bähler et al, 1998) or by crossing with previously created mutant haploids. *rad54Δ* was created using pFA6a-hphMX6 as a template and primer pairs 881–882. *rad52Δ*, *cds1Δ*, and *chk1Δ* were created using pFA6a-natMX6 and primer pairs 772–773, 2,279–2,280, and 2,281–2,282, respectively. *exo1Δ*, *rrp1Δ*, *rev1Δ*, and *srs2Δ* were created using pFA6a-kanMX6 and primer pairs 2,132–2,133, 877–878, 2,673–2,674, and 2,669–2,670, respectively. *rad52-mCherry* and *ssb1-mCherry* were created using primers 1,318–1,319 and 1,236-1,235, respectively, with pGT34 as a template, which carries the mCherry2 coding sequence (Ai et al, 2007), the *adh1* terminator, and the *kanMX6* marker (Afshar et al, 2021). *rad51Δ::kanMX6*, *sfr1Δ::kanMX6*, *rad57Δ::hphMX6 rqh1Δ::natMX6* and *fbh1Δ::kanMX6* have been previously described (Tsutsui et al, 2014; Argunhan et al, 2020; Afshar et al, 2021). For *mCherry-pcn1*, pAV0785 was used (Vještica et al, 2020). For *sad1-*

*mCherry*, a PCR amplicon generated using genomic DNA of MY7270 (Nakazawa et al, 2008) as template with primers 2,144–2,145 was used to transform target strains. *cds1-9xPK* and *chk1-9xPK* were created using primers 2,271–2,272 and 2,275–2,276, respectively, with pNX3c-PK9 as the template (Amelina et al, 2016).

## Media and growth conditions

Standard media and conditions were used for growth (YES), selection (YES with drugs or EMM), and sporulation (SPA) (Hentges et al, 2005). For live-cell imaging, YES media was filter-sterilized.

## Construction of the strain expressing Rad51-mNG

The 500-bp upstream and 500-bp downstream sequences of the *his3⁺* coding sequence were amplified by PCR using genomic DNA as a template with primer pairs 1,014–1,015 and 1,016–1,017, respectively. These two amplicons were combined with the NotI-HindIII-digested pBluescript KSII (+) to obtain pGT12. The 500-bp upstream sequence of the *rad51⁺* gene was PCR-amplified using genomic DNA as a template with primers 1,018–1,019. The fragment carrying the *adh1* terminator and *kanMX6* was PCR-amplified using pNA46 (Afshar et al, 2021) and primers 1,020–1,021. These two amplicons were combined with NotI-HindIII-digested pBluescript KSII (+) to obtain pGT14. A fragment containing the *rad51* promoter, *adh1* terminator and kanMX6 was amplified by PCR using pGT14 as the template with primers 1,027–1,028, which was then cloned at the SmaI site of pGT12 to obtain pGT19. The *rad51⁺* gene was amplified by PCR using genomic DNA as a template with primers 1,027–1,043. A PCR-derived amplicon encoding mNG (Shaner et al, 2013) was generated using primers 1,024–1,030. These two PCR fragments were cloned in tandem at the SmaI site of pGT19 so that the region encoding mNG is fused to the C-terminus of the Rad51 gene, yielding pGT29. All constructs were validated by sequencing. The yeast strain producing Rad51-mNG was generated by digesting pGT29 with NotI and integrating the resulting fragment at the *his3⁺* locus to obtain GT295. The *rad54Δ* derivative (GT325) was obtained by genetic cross.

## Construction of PenotetS overexpression plasmids and related yeast strains

Overexpression plasmids for Rad54, Rrp1, Fbh1, and Srs2 were generated as follows. Coding sequences for Rad54 and Rrp1 were amplified by PCR using genomic DNA as the template with primers 2,690–2,691 (Rad54) and 2,692–2,693 (Rrp1). For Fbh1, which contains an intron, only exon regions were amplified separately using primers 2,696–2,697 (exon 1) and 2,698–2,699 (exon 2). For Srs2, due to the presence of multiple introns, cDNA was synthesized and PCR amplification with primers 2,694–2,695.

The PenotetS vector (pDB5318) was linearized by digestion with NheI and BglII (Lyu et al, 2024). Each PCR-amplified insert was subsequently inserted between the NheI and BglII sites to yield the respective overexpression plasmids: pGT68 (Rad54), pGT70 (Rrp1), pGT74 (Fbh1), and pGT72 (Srs2). These plasmids were linearized

with NotI, then integrated at the $ade6^+$ locus, yielding yeast strains GT1637 (Rad54), GT1641 (Rrp1), GT1647 (Fbh1), and GT1644 (Srs2). The $rad54\Delta$ derivatives of the overexpression strains (GT1669 for Rad54, GT1679 for Rrp1, GT1698 for Fbh1, and GT1687 for Srs2) were obtained by genetic crosses.

## Measurement of cell growth

Exponentially growing cells were diluted to achieve an $OD_{600}$ of 0.04 in 5 ml of YES. The culture was incubated in a 5 ml-capacity, L-shaped tube within a compact rocking incubator (TVS062CA; Advantec) set at 30°C and 70 rpm rocking. Culture optical densities were automatically recorded at 10-min intervals using the Advantec TCS062CA communications software (Ver. 100103). Optical density data were analyzed using the R programming language (Ver 3.5.2). Culture parameters were estimated using the grofit package (Ver. 1.1.1-1) (Rheinahrcampus et al, 2010). Data were processed for figures using ggplot2 (version 3.4.2); in these plots, mean optical densities are shown with SD represented in shaded regions. $\mu_{\log}$ values are presented as dot-plots with mean SD represented by bars. The data were derived from a minimum of six independent experiments (shown individually as points).

## Measurement of cell viability

Fresh cultures were diluted and spotted onto YES plates. For tetrads, 96 cells were picked up and placed on a grid on YES using a tetrad dissector (Singer) per session, followed by a 4-d incubation at 30°C. The viability rate was measured with the number of colonies divided by the number of cells examined.

## Measurement of cell length

Images of cells were captured using a wide field fluorescent microscope (Nikon Eclipse 80i) with a 100x objective fitted with an sCMOS camera (C13440; Hamamatsu). The Cellpose software (version 2.2.2) was used to develop a custom model for segmenting DIC images of fission yeast cells (Stringer et al, 2021). Cell length was quantified using Feret's diameter as a metric, with measurements conducted in the ImageJ software (Schindelin et al, 2012).

## Isolation of the *rad54* temperature-sensitive allele

The fragment carrying the $rad54^+$ gene, including the upstream and downstream sequences (390 and 381 bp, respectively), and a further downstream sequence of the $rad54^+$ gene (465 bp from Chromosome I 3680206 to 3679741), were PCR-amplified using genomic DNA as a template with primer pairs 1,562–1,563 and 1,564–1,565, respectively. The *natMX6* fragment was PCR-amplified using pFA6a-natMX6 with primers 1,564–1,565 (Bähler et al, 1998). These three amplicons were cloned in the order of $rad54^+$ (with upstream and downstream sequences), *natMX6,* and the downstream sequence of $rad54^+$ (465 bp) at the HindIII-NotI sites of pBlueScript KSII (+) to obtain pGT47. Subsequently, pGT47 was used as a template with primers 1,562–1,567 for PCR amplification

using Dream Taq polymerase (Thermo Fisher Scientific), which is inherently mutagenic. The amplicon was introduced into WT cells (GT414), and nourseothricin-resistant colonies were selected at 26°C. The plate carrying transformants was replica-plated onto two YE plates carrying HU at 7.5 mM, one of which was incubated at 26°C while the other at 33°C. Colonies showing sensitivity to HU only at 33°C were selected. The clone showing the strongest temperature dependence was chosen for further characterization (GT636). The $rad54^+$ locus of GT636 had 4 mutations: T1139C, T1479C, T1591, and A1887G, with A of the first ATG as the first nucleotide. These mutations lead to two amino-acid substitutions: V380A and S531P.

## Immunofluorescence analysis

Immunostaining for Rad51 was conducted as previously described (Hagan & Hyams, 1988; Hagan, 2016). Rabbit anti-Rad51 antibody was raised in our previous work (Akamatsu et al, 2003). Cells were fixed by adding a freshly prepared prewarmed 30% formaldehyde solution in PEM (100 mM piperazine-N,N-bis(2-ethanesulfonic acid) [PIPES and sodium salt], 1 mM EGTA, and 1 mM $MgSO_4$, pH 6.9) to the culture, reaching a final concentration of 3.75%. After 45 s, glutaraldehyde was introduced to achieve a final concentration of 0.2%. After 1 h of fixation and three washes in PEM, cells were resuspended in PEMS (PEM and 1.2 M sorbitol) with zymolyase 100-T in PEMS. The cells underwent permeabilization in PEM, resuspension in 1% Triton X-100, and three subsequent washes in PEM. After this, cells were resuspended in PEM with 1 mg/ml $NaBH_3$ for 5 min, followed by two additional washes in PEM. Cells were then resuspended in PEMBAL (PEM + 1% BSA [globulin-free], 0.1% $NaN_3$, and 100 mM lysine HCl) for 30 min. After pelleting, cells were resuspended in PEMBAL with Rad51 antibody at a concentration of 1:300. Incubation overnight at 4°C with agitation was followed by three washes in PEMBAL. Cells were then resuspended in PEMBAL containing appropriate secondary antibodies (Alexa488 or Alexa 594-conjugated anti-rabbit antibodies, Thermo Fisher Scientific). After an additional 3-h incubation at room temperature with agitation, cells were washed three times in PEMBAL and then once in PBS 0.1% $NaN_3$. Finally, cells were resuspended in PBS 0.1% $NaN_3$ with 0.2 µg/ml DAPI.

Imaging was performed on an FV3000 confocal laser-scanning microscope equipped with a 60X/1.30 NA UPlanSApo (Olympus), or an IXplore SpinSR confocal microscope equipped with a 100X/ 1.50 NA UPLanApo-OHR (Olympus) using Fluoview and cellSens Dimension acquisition software, respectively. Representative images are maximum intensity projections of Z-stacks processed by ImageJ (Schindelin et al, 2012). Filament lengths of Rad51 fibers were manually determined. Rad51 fibers less than 2.5 µm in length were categorized as "short fibers," while longer fibers were categorized as "long fibers."

## Live-cell imaging

Suspensions of exponentially growing cells were prepared at a concentration of $1 \times 10^6$ cells/ml in YES. The suspension was loaded into a CellASIC ONIX microfluidic plate (Y04C; Millipore) through cycles of 55 kPa for 5 s and 0 kPa for 5 s. After cell loading, cells were

allowed to settle and stabilize in place by perfusing media through the chamber at 10 kPa for 30 min. For experiments involving *rad54-ts* mutants, cell loading was performed 26°C, followed by stabilization at 33°C. For drug treatment experiments (Fig S15), YE containing bleomycin (1 μg/ml) or HU (20 mM) was introduced at 55 kPa for 5 min after 60 min of incubation with YES at 10 kPa, followed by incubation at 10 kPa with YES carrying each drug. For transient treatment with bleomycin (Fig S16), 60 min of incubation with YES was followed by YES containing bleomycin (1 μg/ml) introduced at 55 kPa for 5 min, with subsequent perfusion at 10 kPa for another 5 min. Following this, drug-free YES was reintroduced to the chamber at 55 kPa for 5 min to wash out bleomycin, with drug-free media continuously perfused thereafter at 10 kPa. Imaging was conducted using the IXplore SpinSR system (Olympus). Unless otherwise indicated, the medium perfusion rate was 10 kPa. Laser power was set at 5%, with exposure ranging from 100 to 250 msec. Binning of 2 × 2 was used for imaging of Sad1-mCherry. Z-stacks comprising 11 slices at 0.5 μm intervals were obtained in these experiments. The interval for time-lapse imaging varied from 1 to 10 min. Quantification was performed using Imaris (Oxford Instruments), wherein the "surface" function was applied to 3D images to define the region of interest (ROI). Consistent surface settings were applied when comparing specimens of different genotypes (e.g., WT versus *rad54* mutant). Raw images were used to quantify ROI signals. Results were processed and visualized using the R programming language. Representative images featured in the figure panels were deconvolved using cellSens Dimension (Olympus).

### Western blotting

Whole cell extracts were prepared by TCA precipitation using $1.0 \times 10^8$ cells per sample from log-phase cultures. Samples were subjected to 10% SDS–PAGE (acrylamide:bis ratio of 99:1) at 150 V for 2 h at 4°C, followed by transfer to PVDF membrane (Immobilon-P, Merck) overnight at 30 V and 4°C. Primary antibodies used were: anti-V5 (1/5,000, mouse monoclonal, MCA1360; Bio-Rad) and anti-tubulin (1/5,000, mouse monoclonal; Sigma-Aldrich). HRP-conjugated anti-mouse IgG (1/5,000; GE Healthcare) was used the secondary antibody. Imaging was performed using a LAS4000 system.

### Random spore analysis

Random spore analysis was performed following a previously described method (Escorcia & Forsburg, 2018). Two parental strains of opposite mating types (*h+* and *h−*) were mixed and spotted onto SPA4S sporulation plates, which were incubated at 26°C for 2 d. After incubation, cells were scraped from the plates and treated with Zymolyase-100T, followed by incubation at 26°C for 6 h. After confirming under a microscope that non-spore cells were completely digested, the suspension was sonicated for 5 min, then plated onto YE5S plates and incubated at 26°C for 4 d. Colonies were then replica plated onto four types of YE5S plates: one containing both G418 and NAT, one with G418 alone, one with NAT alone, and one without any drugs. These replica plates were incubated at 26°C for 1 d, and colony numbers were counted. The fraction (%) of resistant cells was calculated by dividing the colony number on each drug-containing plate by that on the drug-free plate.

The *srs2+* gene is located on chromosome I at coordinates 3,832,775–3,835,593, whereas *rad54+* is located on chromosome I at coordinates 3,683,145–3,680,587, ~150 kb away from *srs2+*. Based on the average genetic distance of 1.7 cM per 10 kb on chromosome I in *S. pombe*, the genetic distance between *rad54+* and *srs2+* is estimated to be ~25.5 cM (Young et al, 2002). The *cds1+* allele is located on chromosome III at coordinates 741,230–739,402, and its segregation is expected to be independent of *srs2+*.

The number of colonies sensitive to both drugs is likely an underestimate. In order for a colony to be scored as drug-sensitive, it must be clonal and well isolated from neighboring colonies. However, after sonication, spores are often not completely separated and tend to remain aggregated, frequently forming mixed-colony clusters. In contrast, colonies resistant to both drugs can be identified more unambiguously, as they are those that grow on YE5S plates containing both drugs.

In the cross between *rad54-ts-NAT* and *srs2Δ::KAN*, 44 colonies (out of 552) were scored as sensitive to both G418 and NAT, whereas no colonies grew on YE5S plates containing both drugs, strongly suggesting that spores inheriting both *rad54-ts-NAT* and *srs2Δ::KAN* alleles were inviable.

### Statistical analyses

All statistical analyses were conducted using the R programming language (Ver 4.1.3). All statistical tests conducted in this study are two-tailed.

# Data Availability

All source data are provided in the supplemental Source Data file.

# Supplementary Information

# Acknowledgements

We thank all members of the Iwasaki laboratory for stimulating discussions and Frank Uhlmann, Jim Haber, Tim Nelson and Beth Rockmill for critical reading of the manuscript. We also thank the National BioResource Project for yeast strains and plasmids, the Fujita lab for sharing their microscopes, and the Cell Biology Center core facility for providing live-cell imaging equipment. We also thank Bilge Argunhan for supervising G Taniguchi at the early stage of his studentship and Haruka Oda for instruction on data processing by Imaris and Yuji Masuda for helpful discussions. This work was supported in part by Grants-in-Aid for Scientific Research (B) (JP18H02371 and JP23H02409 to H Tsubouchi) and Scientific Research (A) (JP18H03985 and JP22H00404 to H Iwasaki) from the Japan Society for the Promotion of Science (JSPS), and by the Basic Research Grant from the Takeda Science Foundation (H Tsubouchi).

## Author Contributions

G Taniguchi: formal analysis, investigation, methodology, and writing—review and editing.
AI May: methodology and writing—review and editing.
H Iwasaki: supervision, funding acquisition, investigation, and writing—review and editing.
H Tsubouchi: conceptualization, supervision, funding acquisition, investigation, project administration, and writing—original draft, review, and editing.

## Conflict of Interest Statement

The authors declare that they have no conflict of interest.

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
