## [Reviewer comments · Life Science Alliance]

Life Science Alliance

Fission yeast Rad54 prevents intergenerational buildup of Rad51 aggregates in proliferating cells

Goki Taniguchi, Alexander May, Hiroshi Iwasaki, and Hideo Tsubouchi

DOI: <https://doi.org/10.26508/lsa.202503252>

Corresponding author(s): *Hideo Tsubouchi, Institute of Science Tokyo and Hiroshi Iwasaki, Tokyo Institute of Technology*

Review Timeline:

Submission Date:	2025-02-05
Editorial Decision:	2025-03-11
Revision Received:	2025-06-21
Editorial Decision:	2025-07-25
Revision Received:	2025-07-30
Accepted:	2025-08-01

Scientific Editor: *Sarita Hebbar*

Transaction Report:

March 11, 2025

Re: Life Science Alliance manuscript #LSA-2025-03252-T

Dr. Hideo Tsubouchi
Tokyo Institute of Technology
Institute of Innovative Research
4259 Nagatsuta, Midori-ku
Yokohama, Kanagawa 226-8503
Japan

Dear Dr. Tsubouchi,

Thank you for submitting your manuscript entitled "Fission yeast Rad54 prevents intergenerational buildup of Rad51 aggregates in proliferating cells" to Life Science Alliance. The manuscript was assessed by expert reviewers, whose comments are appended to this letter.

Overall, the reviewers expressed interest in the proposed role of Rad54 in preventing the accumulation of Rad51 in normally proliferating cells. However, they also raised some concerns that need to be addressed for publication here.

Addressing some of these concerns might require new experimental data to be included and discussed:

- Investigation of the contribution of RRp1, Srs2, Fbh1 in mitigating Rad51 accumulation (Reviewers 2, comments 1 & 2 and Reviewer 3, comments 1 & 2)
- Assessment of the difference in viability/lethality of rad51 and rad54 mutants (Reviewer 3, comments 3 & 5)

Moreover, they have also included suggestions to expand on the discussion with regards to the described interactions (Reviewer 1) and toxicity of the Rad51 aggregates (Reviewer 3, comment 4).

Given these positive recommendations, we would like to invite you to submit a revised manuscript addressing the Reviewer comments. Please include a letter addressing the reviewers' comments point by point.

Thank you for this interesting contribution to Life Science Alliance. We are looking forward to receiving your revised manuscript.

Sincerely,

Sarita Hebbar, PhD
Scientific Editor
Life Science Alliance
<http://www.lsajournal.org>

B. MANUSCRIPT ORGANIZATION AND FORMATTING:

Reviewer #1 (Comments to the Authors (Required)):

This interesting manuscript proposes a novel role for the Rad54 helicase in fission yeast, in opposing accumulation of Rad51 in the absence of external DNA damage. Revisiting the phenotype of *rad54Δ*, they show evidence that the cells have impaired growth and activated DNA damage checkpoint. Numerous reports suggest that a C-terminally tagged Rad51 is likely inactive for protein function but here, they show by co-expression that it accumulates similar to the native protein in a *rad51+* wild type background. This allows quantitative data for fluorescence accumulation.

Overall, the paper is well presented and provides new information of considerable interest.

Only minor questions. First, their data suggest Chk1 is constitutively active, but the cells appear to be able to survive without it. No growth data or plating efficiency is required to see what the effect is on overall growth and survival but their data suggest that the loss of Chk1 inhibits the accumulation of the Rad51 foci. They should discuss this.

Likewise, Exo1 prevents Rad51 accumulation, what are the effects on growth? If the accumulation of Rad51 is responsible for the growth defect, does Exo1 rescue that phenotype?

The data with RPA co-localization are important. These should be included in the main text. A question that is not addressed is whether RPA foci occur independent of Rad51 colocalization. Additionally, we would expect that increased RPA still occurs in *rad51* mutants.

Reviewer #2 (Comments to the Authors (Required)):

Review of the manuscript by Taniguchi et al. entitled "Fission yeast Rad54 prevents intergenerational buildup of RAD51 aggregates in normally proliferating cells"

The aim of this study is to elucidate the mechanism by which Rad54 regulates Rad51 function. The authors use a fission yeast Rad54 mutant with compromised function, leading to a robust accumulation of Rad51 in vegetatively growing cells. The formation of large Rad51 aggregates is frequently associated with cell cycle arrest. A study in budding yeast published in 2011 demonstrated that in a triple translocase mutant (*Rad54, Rdh54, Uls1*), Rad51 accumulates in the nucleus as toxic complexes, leading to genome instability and chromosome loss. This study expands upon those findings using *S. pombe*, where *Rdh54* functions exclusively in meiosis. This system enables the authors to conduct extensive and detailed microscopic analyses. The most compelling aspect of the manuscript involves experiments tracking Rad51 accumulation during cell growth phases, revealing that these aggregates lead to cell cycle arrest in subsequent generations. The study shows that Rad51 foci emerge in

late S phase and depend on the long-range resection machinery. In these conditions, Rad51 is accumulated on ssDNA alongside RPA, resulting in constitutive activation of the DNA damage checkpoint. It is very nice story but would benefit with addressing some of the major points.

Major points:

- 1) A similar study by Muraszko et al. (2021) demonstrated that Rrp1 removes Rad51 from ssDNA. It would be interesting to test whether Rrp1 as well as Rad54 overexpression could reverse the effects of Rad54 deletion or the temperature-sensitive mutant phenotype.
- 2) On the other hand, can Srs2 or Fbh1 overexpression potentially mitigate Rad51 accumulation? It is also possible that the precise timing of these factors' activity would be crucial for their effectiveness.
- 3) The authors speculate that deleting Rrp1 in a Rad54 mutant background would further reduce cell survival. Testing this hypothesis could provide insights into the redundancy of their functions.

Minor points:

1) Typographical and Formatting Issues

- Page 4, Line 15: "uses" → "used"
 - Page 5, Line 12: "We find" → "we have found"
 - Page 7, Line 23: Missing space before "While"
 - Page 10, Lines 14 and 16: "S6b" → "S6B"; "S6a" → "S6A" (this formatting error appears on multiple pages)
 - Page 20, Line 41: Missing space before "1562"
 - Page 29, Line 5: Missing space before "Fig. S7"
- 2) There are several literature citations presented as preprints without volume and page numbers. These references are several years old-please ensure they are properly formatted.
- Page 32, Line 26
 - Page 34, Lines 23, 24, 26
 - Page 35, Lines 6, 38
 - Page 36, Line 6
- 3) Additionally, species names in citations lack italicization throughout the manuscript. There are three instances where the number "(1979)" appears immediately after Science:
- Page 33, Line 20
 - Page 35, Line 22
 - Page 36, Line 9
- 4) Please ensure correct citation formatting.
- 5) Why figures 2B and 3D have different styles?
- 6) What is the extra color (purple?) in figure 4E?
- 7) It would be more informative to show RPA alone, Rad51 alone, and RPA/Rad51 colocalization in Figure S9C.

Reviewer #3 (Comments to the Authors (Required)):

The manuscript "Fission yeast Rad54 prevents intergenerational buildup of Rad51 aggregates in normally proliferating cells" by Taniguchi et al. explores the cellular consequences of RAD54 defects in *S. pombe*. Notably, they isolated a thermosensitive rad54 mutant and could visualize Rad51 structures by immunostaining in fixed wild-type cells and with a recessive Rad51-mNG allele in live cells. They reveal the presence of large and complex Rad51 structures, extending to *S. pombe* recent observations made with bacterial RecA (Wiktor .. Elf Nature 2021, Chimthanawala .. Badrinarayanan, PNAS 2022), *S. cerevisiae* Rad51 (Liu .. Taddei NSMB 2023) and human RAD51 (Sharma, Curr Biol 2024). They provide evidence that these structures originate from frequent replicative DNA lesions, and revealed their inter-generational transmission. Overall the experiments are compelling and well-controlled, the data nicely presented, and the manuscript clearly and logically written. However, certain evidences are circumstantial and several aspects of the mechanism of Rad51 structure formation, growth, and persistence as well as the cause of lethality in the absence of Rad54 are only partially defined. Below I propose simple experiments to address these important points:

1. Nature of the Rad51 aggregates

A central unaddressed point of the paper pertains to the nature of the Rad51 aggregates in Rad54-deficient cells: are they present on ssDNA or dsDNA (or both)? The mutants analyzed (resection, rad52 & rad57 in Fig 1 and 7) loose Rad51 foci formation altogether. It shows that the initiation of the formation of the Rad51 structures requires ssDNA, but does not indicate whether their progressive growth into large aggregates does. Fig S9 shows that only a third of aggregates can be stained with RPA along their length, while the remainder of the cells show a discrete RPA foci. This raises the possibility that Rad51 aggregates grow on dsDNA from a ssDNA seed. The limitation in the data and this possibility are not clearly discussed. It is particularly important because Rad51 filaments on dsDNA represent the only possible substrate for Rad54, a dsDNA (and not ssDNA) translocase.

2. Role of helicases disrupting Rad51 filaments on ssDNA.

Related to the first point: Rad54 disrupts Rad51 filaments formed on dsDNA, and various helicases disrupt Rad51 filaments on ssDNA (Srs2 in budding yeast, PARI, FBH1, FIGNL1, ... in human). The data reported here beg the question: what is the

contribution of Srs2 and Fbh1 in preventing aggregate formation? The srs2 rad54 double-mutant is notoriously inviable, presumably because the forward and backward reactions from a the Rad51 filaments are prevented (see Zinovyev et al., PLoS Comp Biol 2013). The rad54-ts mutant represent a great opportunity to combine it with a deletion of srs2 and observe the consequences on Rad51 filament structure formation. The genetic interaction with fbh1 should also be addressed.

3. Nature of the initiating lesion in S-phase, and cause of the difference in viability of rad51 and rad54 mutants.

The evidence presented here suggest that a recombinogenic substrate forms at every cell cycle in late S-phase. This substrate resolves quickly in WT cells, but persist in the absence of Rad54, leading to an activation of the DNA damage checkpoint and a prolonged cell cycle arrest. Yet, despite the frequent formation of such substrates, rad51-deficient cells are viable, suggesting the existence of a bypass mechanism. In contrast, Rad54-deficient cells loose viability and exhibit Rad51-dependent extended checkpoint arrest, indicating that the backup pathway is blocked in this context. In order to address whether TLS repair the damage in rad51 mutants, and whether the lack of Rad54 prevents TLS access, mutant combination of the rad51, rad54 and rad51rad54 mutant with the main TLS of pombe should be performed. If the rad54 mutant prevents usage of the backup TLS pathway, it is predicted that the viability and cell cycle delay should be equivalent in the rad54 mutant, the rad51-tls double-mutant, and the rad51-rad54-tls triple mutant. It would allow the author to advance a more mechanistic model for the loss of viability specifically observed in a rad54 mutant.

4. Toxicity of the Rad51 structure: it appears from the triple rad54 rqh1 exo1 mutant that lacks extended Rad51 structures is not sufficient to rescue the prolonged G2/M arrest. Consequently, the presumed unrepaired damage, and/or minor Rad51 accumulation, is sufficient to prevent division. I think this fact should be highlighted in the discussion.

5. Cause of lethality in the rad54 mutant. Based on the visualization of chromatin bridges and unequal chromosome mass in Fig. 4G, authors suggest that chromosome segregation errors in the absence of Rad54 are causative of lethality. However, the frequency of such events (~10%) is much lower than the lethality in rad54D cells (~75% Fig. 2G). I would suggest two things to clarify this point. First, perform a chromosome mis-segregation assay using the ade6 hetero-allele system to better evaluate the actual aneuploidy rate. Second, compare chromosome mis-segregation in the rad51, rad54 and double mutant using this system. This should help better define the nature of the lethal event specifically affecting rad54 cells.

Minor comments:

- Figure 3A: No statistical information is provided for WT, cds1 Δ and chk1 Δ cell length values.
- Figure 3B: The authors should specify the conditions of HU and Bleomycin treatment applied before cell harvesting.
- Page 8 Line 34: The sentence refers to Fig. S3B and not S3D.
- Page 9 Line 18-19: The sentences refer to Fig. 4D and 4E instead of 3D and 3E, respectively.
- The authors could cite PMID: 36544019 when discussing the intergenerational transmission of DNA damage

We are very grateful to the reviewers for their constructive comments. Below, the original comments by the editor and reviewers are reproduced in blue italics, and our responses are provided in black text. Any references to page/line numbers to indicate parts of the manuscript that have been modified are accurate for the revised, marked-up version of the manuscript. Major changes made to the manuscript are indicated in red throughout the manuscript.

Dear Dr. Tsubouchi,

Thank you for submitting your manuscript entitled "Fission yeast Rad54 prevents intergenerational buildup of Rad51 aggregates in proliferating cells" to Life Science Alliance. The manuscript was assessed by expert reviewers, whose comments are appended to this letter.

Overall, the reviewers expressed interest in the proposed role of Rad54 in preventing the accumulation of Rad51 in normally proliferating cells. However, they also raised some concerns that need to be addressed for publication here. Addressing some of these concerns might require new experimental data to be included and discussed:

- Investigation of the contribution of RRp1, Srs2, Fbh1 in mitigating Rad51 accumulation (Reviewers 2, comments 1& 2 and Reviewer 3, comments 1& 2)*
- Assessment of the difference in viability/lethality of rad51 and rad54 mutants (Reviewer 3, comments 3 & 5)*

Moreover, they have also included suggestions to expand on the discussion with regards to the described interactions (Reviewer 1) and toxicity of the Rad51 aggregates (Reviewer 3, comment 4).

Given these positive recommendations, we would like to invite you to submit a revised manuscript addressing the Reviewer comments. Please include a letter addressing the reviewers' comments point by point.

Thank you for this interesting contribution to Life Science Alliance. We are looking forward to receiving your revised manuscript.

Sincerely,

*Sarita Hebbar, PhD
Scientific Editor
Life Science Alliance
<http://www.lsjournal.org>*

Reviewer #1 (Comments to the Authors (Required)):

*This interesting manuscript proposes a novel role for the Rad54 helicase in fission yeast, in opposing accumulation of Rad51 in the absence of external DNA damage. Revisiting the phenotype of *rad54Δ*, they show evidence that the cells have impaired growth and activated DNA damage checkpoint. Numerous reports suggest that a C-terminally tagged Rad51 is likely inactive for protein function but here, they show by co-expression that it accumulates similar to the native protein in a *rad51+* wild type background. This allows quantitative data for fluorescence accumulation.*

Overall, the paper is well presented and provides new information of considerable interest..

We thank reviewer #1 for their interest in our work and positive assessment of our manuscript.

1. Only minor questions. First, their data suggest Chk1 is constitutively active, but the cells appear to be able to survive without it. No growth data or plating efficiency is required to see what the effect is on overall growth and survival but their data suggest that the loss of Chk1 inhibits the accumulation of the Rad51 foci. They should discuss this.

2. Likewise, Exo1 prevents Rad51 accumulation, what are the effects on growth? If the accumulation of Rad51 is responsible for the growth defect, does Exo1 rescue that phenotype?

Thank you for suggesting these experiments. Since the two comments above are closely related, we have provided a combined response to them below. We also note that a related issue was raised by reviewer #3 (comment 4).

In light of the reviewer's constructive comments, we have added new data showing that both the *chk1 rad54* and *exo1 rad54* double mutants exhibit slower growth than the *rad54* single mutant (Figs. S1B and S13). In addition, we have revised the discussion to address why certain mutations that reduce Rad51 accumulation exacerbate the growth defect in the *rad54* mutant.

We speculate that in *chk1 rad54* double mutant cells the combined defects in homologous recombination and the DNA damage checkpoint may outweigh the modest benefit conferred by reduced Rad51 accumulation. Similarly, in the *exo1 rad54* double mutant, checkpoint activation via the short-range nuclease, together with a possible impairment in HR-independent functions of the long-range exonucleases, may also outweigh the benefit of reduced Rad51 accumulation.

A detailed discussion has been added as indicated below (page 19, line 17–):

Rad51 accumulation and the cell cycle progression

Live-cell imaging of Rad51 revealed a close correlation between the intensity of Rad51 signals and the duration of the G2 phase. Given that Rad51 accumulation is triggered by DNA breaks and Rad51 is recruited to damaged DNA, it is likely that the DNA damage checkpoint is activated in a subset of cells attempting to repair such damage. This idea is supported by the sustained activation of Chk1 kinase in the *rad54* mutant, where Rad51 accumulation is prominent (Fig 3B). In the absence of Chk1, cell length is reduced to that of wild-type cells. Similarly, Rad51 levels are much lower than in the *rad54* single mutant, although still higher than in wild type. These observations indicate that DNA damage, accompanied by Rad51 recruitment, causes cell cycle delay via DNA damage checkpoint activation, and that Rad51 accumulation occurs gradually in cells characterized by slowed cell cycle progression.

As Rad51 accumulation inhibits cell growth, the reduced Rad51 accumulation observed in the *rad54 chk1* double mutant cells might be expected to allow improve cell growth. However, we observed that colony size is much smaller than in the *rad54* single mutant colonies (Fig S1B), indicating that the *chk1* mutation exacerbates the growth defect caused by *rad54*. This is likely due to a combination of HR defects and impaired DNA damage checkpoint function, resulting in attempted mitosis with unrepaired DNA. The deleterious effects of checkpoint failure therefore appear to outweigh the modest benefit of reduced Rad51 accumulation.

Similarly, under conditions in which long-range exonuclease activity is

compromised (i.e., in the *exo1 rqh1 rad54* triple mutant), Rad51 accumulation is substantially reduced, presumably due to limited resection (Fig. 7D). Despite this reduction, the cell cycle remains delayed, indicating persistent checkpoint activation. These observations suggest that short-range resection activity provided by the MRN–Ctp1 complex is sufficient to sustain checkpoint signaling. Furthermore, long-range exonuclease activity may contribute to pathways independent of HR, and its loss could also negatively impact cell growth. We speculate that these negative effects outweigh the relatively minor benefit conferred by reduced Rad51 accumulation.

3. The data with RPA co-localization are important. These should be included in the main text. A question that is not addressed is whether RPA foci occur independent of Rad51 col-localization. Additionally, we would expect that increased RPA still occurs in rad51 mutants.

We thank reviewer #1 for highlighting the importance of the RPA data. We also note that a similar issue was raised by reviewer #2 (comment 7).

We have now moved RPA colocalization data to the main figures in the revised Fig. 8.

To address the reviewer's question about colocalization, we did not observe RPA foci that did not colocalize with Rad51. We have revised the figure to display the data as bar graphs rather than pie charts to more clearly make this point (Fig. 8).

With respect to RPA localization in the *rad51* mutant, it is well established that RPA accumulates in *rad51* mutants (e.g., Lisby et al., *Cell*). Moreover, the focus of our analysis is to determine whether the accumulated Rad51 in the *rad54* mutant is associated with ssDNA (using RPA as a marker for ssDNA). Therefore, we believe that examining RPA localization in the *rad51* mutant is beyond the scope of the current study.

Reviewer #2 (Comments to the Authors (Required)):

Review of the manuscript by Taniguchi et al. entitled "Fission yeast Rad54 prevents intergenerational buildup of RAD51 aggregates in normally proliferating cells"

The aim of this study is to elucidate the mechanism by which Rad54 regulates Rad51 function. The authors use a fission yeast Rad54 mutant with compromised function, leading to a robust accumulation of Rad51 in vegetatively growing cells. The formation of large Rad51 aggregates is frequently associated with cell cycle arrest. A study in budding yeast published in 2011 demonstrated that in a triple translocase mutant (Rad54, Rdh54, Uls1),

*Rad51 accumulates in the nucleus as toxic complexes, leading to genome instability and chromosome loss. This study expands upon those findings using *S. pombe*, where *Rdh54* functions exclusively in meiosis. This system enables the authors to conduct extensive and detailed microscopic analyses. The most compelling aspect of the manuscript involves experiments tracking *Rad51* accumulation during cell growth phases, revealing that these aggregates lead to cell cycle arrest in subsequent generations. The study shows that *Rad51* foci emerge in late S phase and depend on the long-range resection machinery. In these conditions, *Rad51* is accumulated on ssDNA alongside RPA, resulting in constitutive activation of the DNA damage checkpoint. It is very nice story but would benefit with addressing some of the major points.*

We thank reviewer 2 for their thorough reading of the manuscript and positive evaluation of our work.

Major points:

*1) A similar study by Muraszko et al. (2021) demonstrated that *Rrp1* removes *Rad51* from ssDNA. It would be interesting to test whether *Rrp1* as well as *Rad54* overexpression could reverse the effects of *Rad54* deletion or the temperature-sensitive mutant phenotype.*

*2) On the other hand, can *Srs2* or *Fbh1* overexpression potentially mitigate *Rad51* accumulation? It is also possible that the precise timing of these factors' activity would be crucial for their effectiveness.*

*3) The authors speculate that deleting *Rrp1* in a *Rad54* mutant background would further reduce cell survival. Testing this hypothesis could provide insights into the redundancy of their functions.*

Thank you for providing constructive feedback about the relationship between *Rad51* and *Rrp1*, *Srs2* and *Fbh1*. Since the three comments above are closely related, we have provided a combined response to them below. We also note that related issues were raised by reviewer #3 (comment 2).

To address the reviewer's comments, we investigated the potential functional redundancy between *Rrp1*, *Srs2*, *Fbh1*, and *Rad54* by overexpressing these proteins. As the reviewer hypothesized, these experiments indicate that overexpression of any of these proteins substantially reduces *Rad51* accumulation (Fig. S4A, B). We have described these findings in the revised text at page 7, line 26-.

To further examine the relationship between *Rrp1* and *Rad54*, we also tested whether *Rrp1* plays a role in the *rad54* mutant background by constructing a *rrp1 rad54* double mutant strain. The level of *Rad51* accumulation in the absence of both these proteins was essentially equivalent to that observed in

the *rad54* single mutant. These results suggest that Rrp1 contributes little, if at all, to the Rad54-independent repair pathway (page 7, line 31; Fig. S2A, B).

We have included a section in the discussion that addresses the possible roles of these proteins in Rad54-independent repair mechanisms (page 18, line 8–):

Although *rad54* mutant cells exhibit strong growth defects, about a quarter of these cells remain viable, suggesting that alternative mechanisms allow their survival. We examined a group of proteins involved in Rad51 removal—namely, Srs2, Fbh1, and Rrp1 (Huselid & Bunting, 2020). When overexpressed in *rad54* mutant cells, any of these three proteins substantially reduced Rad51 accumulation. This observation suggests that Srs2, Fbh1, and Rrp1 can negatively regulate Rad51 accumulation. *S. cerevisiae* Srs2 and *S. pombe* Fbh1 are known to displace Rad51 from ssDNA, while *S. pombe* Rrp1 removes Rad51 from dsDNA (Krejci et al, 2003; Veaute et al, 2003; Tsutsui et al, 2014; Muraszko et al, 2021). Beyond direct physical displacement of Rad51, it is also possible that Fbh1 and Rrp1 reduce Rad51 levels by promoting its degradation. Fbh1 contains an F-box motif and functions with the SCF complex as an E3 ubiquitin ligase targeting Rad51 (Tsutsui et al, 2014). Similarly, Rrp1 contains a RING domain and also serves as an E3 ligase for Rad51 (Muraszko et al, 2021).

Rrp1, however, is unlikely to play a major role in the Rad54-independent repair pathway, since the *rrp1 rad54* double mutant is essentially indistinguishable from the *rad54* single mutant. In contrast, the *srs2* null mutation, when combined with *rad54*, is lethal in both *S. cerevisiae* and *S. pombe*, suggesting a close functional relationship between the two genes (Palladino & Klein, 1992; Maftahi et al, 2002; Zinovyev et al, 2013). This synthetic lethality prevented us from examining the effect of Srs2 on Rad51 accumulation in the *rad54* mutant. We therefore attempted to use the *rad54-ts* mutation, but were unable to isolate a *srs2 rad54-ts* double mutant even at the permissive temperature. Since the *rad54-ts* mutant exhibits only a very subtle phenotype under permissive conditions, this finding further supports a strong genetic interaction between *srs2* and *rad54*. Thus, Srs2 likely plays a major role in the Rad54-independent repair pathway. In the *fbh1 rad54* double mutant, the proportion of cells exhibiting Rad51 fibers was markedly reduced, while the fraction of cells with Rad51 signals was comparable to that in the *rad54* mutant. This pattern resembles the characteristic Rad51 localization observed in *fbh1* mutants, where cells with long Rad51 fibers are relatively rare. Given that Rad51 fibers originate from foci, Fbh1 appears to function in converting Rad51 foci into fibers. If the fiber represents a long Rad51 nucleoprotein filament, the requirement for Fbh1 suggests that occasional removal of Rad51 from DNA may be necessary for its continuous extension.

Minor points:

1) Typographical and Formatting Issues

- Page 4, Line 15: "uses" → "used"
- Page 5, Line 12: "We find" → "we have found"
- Page 7, Line 23: Missing space before "While"
- Page 10, Lines 14 and 16: "S6b" → "S6B"; "S6a" → "S6A" (this formatting error appears on multiple pages)
- Page 20, Line 41: Missing space before "1562"
- Page 29, Line 5: Missing space before "Fig. S7"

Thank you very much – we greatly appreciate the reviewer’s careful reading and attention to detail. We have amended the text accordingly in the revised version of the manuscript.

2) There are several literature citations presented as preprints without volume and page numbers. These references are several years old-please ensure they are properly formatted.

- Page 32, Line 26
- Page 34, Lines 23, 24, 26
- Page 35, Lines 6, 38
- Page 36, Line 6

Thank you. These references have now been updated with the appropriate publication details and formatting.

3) Additionally, species names in citations lack italicization throughout the manuscript.

Species names in all citations have been italicized accordingly.

There are three instances where the number "(1979)" appears immediately after Science:

- Page 33, Line 20
- Page 35, Line 22
- Page 36, Line 9

This formatting error has been corrected in the revised manuscript.

4) Please ensure correct citation formatting.

All citation formatting has been reviewed. We believe that the inconsistencies noted by the reviewer have been corrected in the revised manuscript.

5) Why figures 2B and 3D have different styles?

The discrepancy between these figures is due to the different sample sizes of

the experiments. In Fig. 2B, the number of samples is in the thousands, while in Fig. 3D, it is in the hundreds. For the latter, we displayed all individual data points for clarity. We felt that this was not practical for the very large dataset presented in Fig. 2B.

6) What is the extra color (purple?) in figure 4E?

Our apologies if the color scheme in Figure 4E was unclear. The additional color (dark purple) represents the region where the shaded areas—indicating the standard deviations of the two strains—overlap. To clarify this point, we have revised the figure legend as follows:

(E) Kinetics of Rad51 appearance in the time course shown in **(D)**. Solid lines indicate the average of replicates; shaded areas represent the standard deviation. The area where the shaded regions (light blue and gray) overlap appears in a blended color, which reflects the overlap of standard deviations between the two strains.

7) It would be more informative to show RPA alone, Rad51 alone, and RPA/Rad51 colocalization in Figure S9C.

Thank you very much for the suggestion. We note that a similar issue was raised by reviewer #1 (comment 3).

First of all, we note that Rad51 localization manifests as either foci or as foci accompanied by fibers. We found that fibers are observed only in *rad54Δ* mutant cells, suggesting that these structures are functionally distinct. The reviewer suggests that RPA localization can be categorized into three groups: RPA only, Rad51 only, and RPA/Rad51 colocalization. However, we feel that this categorization does not capture the full implications of the data for two key reasons. First, the difference between foci and fiber localization is functionally important and therefore should be quantified. Second, we did not observe any RPA foci forming in the absence of Rad51 foci.

We have therefore provided the new Fig. 8 in which we have adopted reviewer #2's suggested categorization, but have further subdivided RPA-Rad51 colocalization into foci-only and foci/fiber localization. These data are shown as a bar graph in this figure. We thank the reviewer for indicating how we could better present the implication of our data.

Reviewer #3 (Comments to the Authors (Required)):

*The manuscript "Fission yeast Rad54 prevents intergenerational buildup of Rad51 aggregates in normally proliferating cells" by Taniguchi et al. explores the cellular consequences of RAD54 defects in *S. pombe*. Notably, they*

isolated a thermosensitive rad54 mutant and could visualize Rad51 structures by immunostaining in fixed wild-type cells and with a recessive Rad51-mNG allele in live cells. They reveal the presence of large and complex Rad51 structures, extending to S. pombe recent observations made with bacterial RecA (Wiktor .. Elf Nature 2021, Chimthanawala .. Badrinarayanan, PNAS 2022), S. cerevisiae Rad51 (Liu .. Taddei NSMB 2023) and human RAD51 (Sharma, Curr Biol 2024). They provide evidence that these structures originate from frequent replicative DNA lesions, and revealed their inter-generational transmission. Overall the experiments are compelling and well-controlled, the data nicely presented, and the manuscript clearly and logically written. However, certain evidences are circumstantial and several aspects of the mechanism of Rad51 structure formation, growth, and persistence as well as the cause of lethality in the absence of Rad54 are only partially defined. Below I propose simple experiments to address these important points:

We thank reviewer #3 for their positive appraisal of our manuscript in the context of recent work on Rad51.

1. Nature of the Rad51 aggregates

A central unaddressed point of the paper pertains to the nature of the Rad51 aggregates in Rad54-deficient cells: are they present on ssDNA or dsDNA (or both)? The mutants analyzed (resection, rad52 & rad57 in Fig 1 and 7) loose Rad51 foci formation altogether. It shows that the initiation of the formation of the Rad51 structures requires ssDNA, but does not indicate whether their progressive growth into large aggregates does. Fig S9 shows that only a third of aggregates can be stained with RPA along their length, while the remainder of the cells show a discrete RPA foci. This raises the possibility that Rad51 aggregates grow on dsDNA from a ssDNA seed. The limitation in the data and this possibility are not clearly discussed. It is particularly important because Rad51 filaments on dsDNA represent the only possible substrate for Rad54, a dsDNA (and not ssDNA) translocase.

Thank you very much for raising this important point regarding the nature of Rad51 aggregates in *rad54* mutant cells.

We agree with reviewer that our data clearly show that the initiation of Rad51 foci formation depends on ssDNA (supported by the loss of Rad51 structures in resection-deficient mutants). We also appreciate that it remains less certain whether the subsequent growth of these aggregates also occurs on ssDNA or proceeds further into surrounding dsDNA. This distinction is particularly important in light of Rad54's known ability to disassemble Rad51 filaments on dsDNA, but not on ssDNA.

While this point is very difficult to address experimentally, we have added a paragraph at line 33 (page 17) to specifically address this limitation and discuss

the possibility that Rad51 filaments may extend from ssDNA onto dsDNA as they grow. The text of this new paragraph is presented below:

Given that Rad54 functions primarily during the synaptic phase and thereafter, Rad51 aggregates likely represent Rad51 molecules accumulating on gradually exposed ssDNA. This view is consistent with data showing that Rad51 aggregation is suppressed when DNA end resection is compromised. However, it is possible that exposed ssDNA is only required to initiate Rad51 aggregation, and that Rad51 filament formation may extend beyond the ssDNA/dsDNA junction, covering dsDNA to form larger aggregates. Indeed, Rad54 has been shown to remove Rad51 from dsDNA, but not from ssDNA (Solinger et al, 2002b; Ceballos & Heyer, 2011). We observed that only one-third of Rad51 filaments co-localize with RPA, possibly reflecting association of Rad51 with dsDNA. However, this observation might simply reflect the mutually exclusive binding of Rad51 and RPA to ssDNA, as only one of these ssDNA-binding proteins can bind a given region at a time.

2. Role of helicases disrupting Rad51 filaments on ssDNA.

Related to the first point: Rad54 disrupts Rad51 filaments formed on dsDNA, and various helicases disrupt Rad51 filaments on ssDNA (Srs2 in budding yeast, PARI, FBH1, FIGNL1, ... in human). The data reported here beg the question: what is the contribution of Srs2 and Fbh1 in preventing aggregate formation? The srs2 rad54 double-mutant is notoriously inviable, presumably because the forward and backward reactions from a the Rad51 filaments are prevented (see Zinovyev et al., PLoS Comp Biol 2013). The rad54-ts mutant represent a great opportunity to combine it with a deletion of srs2 and observe the consequences on Rad51 filament structure formation. The genetic interaction with fbh1 should also be addressed.

Thank you very much for raising this interesting point. We note that related issues were raised by reviewer #2 (comment 3).

Based on this feedback, we examined whether Fbh1 contributes to the prevention of Rad51 accumulation by constructing a *fbh1 rad54* double mutant strain and looking at Rad51 foci/fiber formation. We found that the proportion of cells exhibiting Rad51 fibers was significantly reduced in the double mutant, while the proportion of cells with Rad51 foci increased. These results suggest that Fbh1 functions in converting Rad51 foci into fibers (page 7, line 37; Fig. S2A, B).

To assess the role of Srs2, we attempted to generate a *rad54-ts srs2Δ* double mutant, as the *rad54Δ srs2Δ* combination is known to be lethal. However, we were unable to recover a *rad54-ts srs2Δ* strain, even at the permissive temperature. This was unexpected given the mild phenotype observed in *rad54-*

ts mutant cells at permissive temperature. This highly sensitive genetic interaction provides further support for a critical functional overlap between Srs2 and Rad54 (page 8, line 6-; Fig. S3).

In addition, we overexpressed Srs2 and Fbh1 in the *rad54* mutant background and observed that both proteins substantially reduce Rad51 accumulation (page 8, line 16-; Fig. S4A, B).

We have incorporated a discussion of the potential roles of Fbh1, Srs2, and also Rrp1 (as raised by reviewer #2) in Rad54-independent pathways (page 18, line 7-), as reproduced below:

The mechanisms that support the viability in the absence of Rad54

Although *rad54* mutant cells exhibit strong growth defects, about a quarter of these cells remain viable, suggesting that alternative mechanisms allow their survival. We examined a group of proteins involved in Rad51 removal—namely, Srs2, Fbh1, and Rrp1 (Huselid & Bunting, 2020). When overexpressed in *rad54* mutant cells, any of these three proteins substantially reduced Rad51 accumulation. This observation suggests that Srs2, Fbh1, and Rrp1 can negatively regulate Rad51 accumulation. *S. cerevisiae* Srs2 and *S. pombe* Fbh1 are known to displace Rad51 from ssDNA, while *S. pombe* Rrp1 removes Rad51 from dsDNA (Krejci et al, 2003; Veaute et al, 2003; Tsutsui et al, 2014; Muraszko et al, 2021). Beyond direct physical displacement of Rad51, it is also possible that Fbh1 and Rrp1 reduce Rad51 levels by promoting its degradation. Fbh1 contains an F-box motif and functions with the SCF complex as an E3 ubiquitin ligase targeting Rad51 (Tsutsui et al, 2014). Similarly, Rrp1 contains a RING domain and also serves as an E3 ligase for Rad51 (Muraszko et al, 2021).

Rrp1, however, is unlikely to play a major role in the Rad54-independent repair pathway, since the *rrp1 rad54* double mutant is essentially indistinguishable from the *rad54* single mutant. In contrast, the *srs2* null mutation, when combined with *rad54*, is lethal in both *S. cerevisiae* and *S. pombe*, suggesting a close functional relationship between the two genes (Palladino & Klein, 1992; Maftahi et al, 2002; Zinovyev et al, 2013). This synthetic lethality prevented us from examining the effect of Srs2 on Rad51 accumulation in the *rad54* mutant. We therefore attempted to use the *rad54-ts* mutation, but were unable to isolate a *srs2 rad54-ts* double mutant even at the permissive temperature. Since the *rad54-ts* mutant exhibits only a very subtle phenotype under permissive conditions, this finding further supports a strong genetic interaction between *srs2* and *rad54*. Thus, Srs2 likely plays a major role in the Rad54-independent repair pathway. In the *fbh1 rad54* double mutant, the proportion of cells exhibiting Rad51 fibers was markedly reduced, while the fraction of cells with Rad51 signals was comparable to that in the *rad54* mutant. This pattern resembles the characteristic Rad51 localization observed in *fbh1* mutants, where cells with long Rad51 fibers are

relatively rare. Given that Rad51 fibers originate from foci, Fbh1 appears to function in converting Rad51 foci into fibers. If the fiber represents a long Rad51 nucleoprotein filament, the requirement for Fbh1 suggests that occasional removal of Rad51 from DNA may be necessary for its continuous extension.

3. Nature of the initiating lesion in S-phase, and cause of the difference in viability of rad51 and rad54 mutants.

The evidence presented here suggest that a recombinogenic substrate forms at every cell cycle in late S-phase. This substrate resolves quickly in WT cells, but persist in the absence of Rad54, leading to an activation of the DNA damage checkpoint and a prolonged cell cycle arrest. Yet, despite the frequent formation of such substrates, rad51-deficient cells are viable, suggesting the existence of a bypass mechanism. In contrast, Rad54-deficient cells loose viability and exhibit Rad51-dependent extended checkpoint arrest, indicating that the backup pathway is blocked in this context. In order to address whether TLS repair the damage in rad51 mutants, and whether the lack of Rad54 prevents TLS access, mutant combination of the rad51, rad54 and rad51rad54 mutant with the main TLS of pombe should be performed. If the rad54 mutant prevents usage of the backup TLS pathway, it is predicted that the viability and cell cycle delay should be equivalent in the rad54 mutant, the rad51-tls double-mutant, and the rad51-rad54-tls triple mutant. It would allow the author to advance a more mechanistic model for the loss of viability specifically observed in a rad54 mutant.

We thank the reviewer for suggesting additional evidence to advance the mechanistic merit of our analyses.

To follow up this interesting suggestion, we examined the potential involvement of translesion synthesis (TLS) in the Rad54-independent repair pathway. Rev1, a Y-family DNA polymerase, and Rev3, the catalytic subunit of Polζ, are major components of the TLS mechanism responsible for most spontaneous mutations. To assess their roles, we introduced *rev1* or *rev3* mutation into *rad51* and *rad54* mutant backgrounds.

Although *rad54* mutant cells show severe growth defects, this phenotype was exacerbated by the introduction of *rev1* or *rev3* mutations. Similarly, strains defective in both homologous recombination and TLS—specifically, the *rev1 rad51*, *rev3 rad51*, *rev1 rad51 rad54*, and *rev3 rad51 rad54* mutants—also exhibited severely compromised growth (Fig. S5).

To indirectly test whether the accumulation of Rad51 in *rad54* mutants interferes with TLS, we compared cell growth between *rev1 rad54* and *rev1 rad51* strains, amongst other strains. However, since *rad51* mutants already exhibit poor growth—and *rad54* mutants even more so—combining either with TLS defects resulted in severely compromised growth in all cases, making it

difficult to draw quantitative conclusions in these experiments.

In addition, these double and triple mutants displayed notable variation in colony size, likely due to increased genome instability, which further complicated any interpretation of subtle growth differences. Therefore, while our results support the involvement of TLS in Rad54-independent repair mechanisms, it remains unclear whether TLS is actively inhibited by the accumulation of Rad51 in the *rad54* mutant.

These findings are discussed on page 8 (from line 34), as below:

HR and the translesion synthesis (TLS) pathway have been shown to play redundant roles in the process of bypassing damaged DNA that otherwise results in the obstruction of replication forks. Therefore, we sought to investigate the role of TLS in Rad54-independent damage repair. Rev1, a member of the Y-family DNA polymerase, and Rev3, the catalytic subunit of Pol ζ , are key components of the TLS pathway that causes the majority of spontaneous mutations (Lawrence, 2004). We combined the *rev1* or *rev3* mutation with the *rad54* null mutation to examine the involvement of TLS in Rad54-independent mechanisms. *rad54* cell growth was further impaired when combined with *rev1* or *rev3*, suggesting that the TLS pathway allows growth when Rad54 is absent (Fig. S5A). TLS mutations caused similar effects in *rad51* or *rad51 rad54* background strains (Fig. S5B). These results suggest that the growth reduction observed in these mutants is not unique to the *rad54* mutation, likely due to the synthetic effects brought by the combination of both HR and TLS defects.

As well as on page 19 (line 1 onwards):

HR and the TLS pathway collaborate to overcome DNA damage that blocks replication forks. Rev1, a Y-family DNA polymerase, and Rev3, the catalytic subunit of Pol ζ , are major players in the TLS mechanism responsible for most spontaneous mutations (Lawrence, 2004). The already severely impaired growth phenotype of *rad54* cells was further exacerbated by introduction of *rev1* or *rev3* mutations. Similar results were observed when these TLS mutations were introduced into *rad51* or *rad51 rad54* double mutants, consistent with the cooperative role of HR and TLS during replication. It is possible that the accumulated Rad51 in *rad54* mutants interferes with TLS activity. However, we found it difficult to draw definitive conclusions about growth differences between strains such as *rev1 rad51* and *rev1 rad54*. This difficulty arises primarily because introducing *rev1* or *rev3* into either the *rad51* or *rad54* mutant background results in extremely impaired growth phenotypes. Furthermore, these double and triple mutants often exhibited notable colony size variation, likely due to increased genome instability, further complicating direct comparisons.

4. Toxicity of the Rad51 structure: it appears from the triple rad54 rqh1 exo1 mutant that lacks extended Rad51 structures is not sufficient to rescue the prolonged G2/M arrest. Consequently, the presumed unrepaired damage, and/or minor Rad51 accumulation, is sufficient to prevent division. I think this fact should be highlighted in the discussion.

Thank you very much for your constructive suggestion. We note that a related suggestion was also made by reviewer #1 (comment 2).

In light of this helpful suggestion, we have added a new section discussing the relationship between Rad51 accumulation and cell cycle progression. In particular, we highlighted the observation that the *rad54 rqh1 exo1* triple mutant, which lacks extended Rad51 structures, still exhibits a prolonged G2 phase (page 19, line 38 onwards). This finding suggests that the presence of unrepaired DNA damage—and/or even a modest level of Rad51 accumulation—is sufficient to trigger and maintain the DNA damage checkpoint, thereby preventing cell division.

Similarly, under conditions in which long-range exonuclease activity is compromised (i.e., in the *exo1 rqh1 rad54* triple mutant), Rad51 accumulation is substantially reduced, presumably due to limited resection (Fig. 7D). Despite this reduction, the cell cycle remains delayed, indicating persistent checkpoint activation. These observations suggest that short-range resection activity provided by the MRN–Ctp1 complex is sufficient to sustain checkpoint signaling. Furthermore, long-range exonuclease activity may contribute to pathways independent of HR, and its loss could also negatively impact cell growth. We speculate that these negative effects outweigh the relatively minor benefit conferred by reduced Rad51 accumulation.

5. Cause of lethality in the rad54 mutant. Based on the visualization of chromatin bridges and unequal chromosome mass in Fig. 4G, authors suggest that chromosome segregation errors in the absence of Rad54 are causative of lethality. However, the frequency of such events (~10%) is much lower than the lethality in rad54D cells (~75% Fig. 2G). I would suggest two things to clarify this point. First, perform a chromosome mis-segregation assay using the ade6 hetero-allele system to better evaluate the actual aneuploidy rate. Second, compare chromosome mis-segregation in the rad51, rad54 and double mutant using this system. This should help better define the nature of the lethal event specifically affecting rad54 cells.

Thank you very much for your thoughtful suggestion.

We would like to clarify that we did not intend to claim that chromosome

segregation errors are a primary cause of lethality in *rad54* mutant cells and regret if the previous version of the manuscript suggested so. We intended to highlight that the majority of nuclear divisions in these cells produce evenly sized daughter nuclei. We therefore agree with the reviewer that chromosome segregation errors are unlikely to fully explain the high lethality observed in the *rad54* mutant.

To avoid any ambiguity regarding this point we have revised the text as follows (page 10, from line 24-):

These findings suggest that accumulated Rad51 and the associated DNA damage may interfere with the chromosome segregation machinery and/or facilitate inter-chromosomal associations. However, the relatively low occurrence of aberrant segregation events (~10%) does not account for the high lethality of the *rad54* mutant (~75%).

Minor comments:

- *Figure 3A: No statistical information is provided for WT, *cds1Δ* and *chk1Δ* cell length values.*

Statistical information has now been included in the figure, in line with the reviewer's suggestion.

- *Figure 3B: The authors should specify the conditions of HU and Bleomycin treatment applied before cell harvesting.*

These experimental conditions have been added to the figure legend.

- *Page 8 Line 34: The sentence refers to Fig. S3B and not S3D.*

This sentence contained a typographical error. The localization assay was conducted using bleomycin-treated wild-type cells, not *rad54* mutants. The sentence has been corrected accordingly, and the figure reference remains unchanged (Fig. S7D, formerly Fig. S3D). Thank you again for your close reading of the manuscript.

- *Page 9 Line 18-19: The sentences refer to Fig. 4D and 4E instead of 3D and 3E, respectively.*

These figure panel references have been corrected to indicate the appropriate figures.

- *The authors could cite PMID: 36544019 when discussing the intergenerational transmission of DNA damage*

Thank you for bringing this relevant study to our attention.

We have incorporated the key findings of this study and discussed their relevance in the context of our work, as follows (page 21, line 24 onwards):

In *C. elegans*, DNA damage that is inherited from the male parent leads to the formation of heterochromatin, thereby impeding the action of homologous recombination mechanisms on damaged DNA (Wang et al, 2023). Similarly, the transmission of damaged DNA associated with Rad51 from mother to daughter may interfere with the genome maintenance mechanisms that operate normally, thereby triggering genome destabilization.

July 25, 2025

RE: Life Science Alliance Manuscript #LSA-2025-03252-TR

Dr. Hideo Tsubouchi
Institute of Science Tokyo
4259 Nagatsuta, Midori-ku
Yokohama, Kanagawa 226-8503
Japan

Dear Dr. Tsubouchi,

Thank you for submitting your revised manuscript entitled "Fission yeast Rad54 prevents intergenerational buildup of Rad51 aggregates in proliferating cells". Your revised manuscript was evaluated by two of the original reviewers. Based on their overall assessment, we would be happy to publish your paper in Life Science Alliance pending final revisions necessary to meet our formatting guidelines.

- Please clarify your data availability statement, "All source data are provided in the supplemental Source Data file". Please confirm that all the source data presented in figures is indeed included in the file named, "Figure Source Data". For example, I didn't find data for Figure 6.
- For Figure 6B, the images representing Rad51 row (at 128 and 132 min) appear to be identical. Kindly confirm these images are correct or make required changes in case of error.
- Please provide details for Rad51 antibody (origin, source, and appropriate citation) in the methods section.
- Please add molecular weight markers to the blots displayed in Figure 3B
- Please upload your main manuscript text as an editable doc file.
- Please upload your Tables in editable .doc or Excel format
- Please upload your main and supplementary figures as single files
- Please add your main, supplementary figure, and table legends to the main manuscript text after the references section.
- Please add callouts for Figures 7C, F;8A-B; S9A-D; S15A-C and S16A-C to your main manuscript text
- The contributions selected for Hiroshi Iwasaki do not qualify them for authorship. Please either update the contributions in our system and in the Author Contributions section of the manuscript, or let us know if the author needs to be removed (and added eventually to the acknowledgment section)
- Please add the X and Bluesky handles of your host institute/organization, as well as your own and/or one of the authors, in our system

A. FINAL FILES:

- An editable version of the final text (.DOC or .DOCX) is needed for copyediting (no PDFs).
- High-resolution figure, supplementary figure and video files uploaded as individual files: See our detailed guidelines for preparing your production-ready images, <https://www.life-science-alliance.org/authors>
- Summary blurb (enter in submission system): A short text summarizing in a single sentence the study (max. 200 characters)

including spaces). This text is used in conjunction with the titles of papers, hence should be informative and complementary to the title. It should describe the context and significance of the findings for a general readership; it should be written in the present tense and refer to the work in the third person. Author names should not be mentioned.

B. MANUSCRIPT ORGANIZATION AND FORMATTING:

Sincerely,

Sarita Hebbar, PhD
Scientific Editor
Life Science Alliance
<http://www.lsajournal.org>

Reviewer #1 (Comments to the Authors (Required)):

This paper shows that *S. pombe* rad54 Δ mutants are significantly sicker than rad51 Δ , and suggest that this reflects accumulation of Rad51 aggregates that disrupt normal replication resolution. The authors have addressed my prior concerns and have added significant amounts of additional data. While largely observational/genetical rather than molecularly mechanistic, this identifies an interesting phenomenon and establishes a solid basis for further studies. I do not have any further concerns.

Reviewer #3 (Comments to the Authors (Required)):

The authors have thoroughly addressed the points I had raised, either with new experiments or with changes to the text. I have no remaining comments and recommend publication.

August 1, 2025

RE: Life Science Alliance Manuscript #LSA-2025-03252-TRR

Dr. Hideo Tsubouchi
Institute of Science Tokyo
4259 Nagatsuta, Midori-ku
Yokohama, Kanagawa 226-8503
Japan

Dear Dr. Tsubouchi,

Thank you for submitting your Research Article entitled "Fission yeast Rad54 prevents intergenerational buildup of Rad51 aggregates in proliferating cells". It is a pleasure to let you know that your manuscript is now accepted for publication in Life Science Alliance. Congratulations on this interesting work.

DISTRIBUTION OF MATERIALS:

Again, congratulations on a very nice paper. I hope you found the review process to be constructive and are pleased with how the manuscript was handled editorially. We look forward to future exciting submissions from your lab.

Sincerely,

Sarita Hebbar, PhD
Scientific Editor
Life Science Alliance
<http://www.lsjournal.org>